# Reliable Decisions with Threshold Calibration

**Roshni Sahoo**
Stanford University
rsahoo@stanford.edu

**Shengjia Zhao**
Stanford University
sjzhao@stanford.edu

**Alyssa Chen**
UTSW Medical Center
alyssa.chen@utsw.edu

**Stefano Ermon**
Stanford University
ermon@stanford.edu

## Abstract

Decision makers rely on probabilistic forecasts to predict the loss of different decision rules before deployment. When the forecasted probabilities match the true frequencies, predicted losses will be accurate. Although perfect forecasts are typically impossible, probabilities can be calibrated to match the true frequencies on average. However, we find that this *average* notion of calibration, which is typically used in practice, does not necessarily guarantee accurate decision loss prediction. Specifically in the regression setting, the loss of threshold decisions, which are decisions based on whether the forecasted outcome falls above or below a cutoff, might not be predicted accurately. We propose a stronger notion of calibration called threshold calibration, which is exactly the condition required to ensure that decision loss is predicted accurately for threshold decisions. We provide an efficient algorithm which takes an uncalibrated forecaster as input and provably outputs a threshold-calibrated forecaster. Our procedure allows downstream decision makers to confidently estimate the loss of any threshold decision under any threshold loss function. Empirically, threshold calibration improves decision loss prediction without compromising on the quality of the decisions in two real-world settings: hospital scheduling decisions and resource allocation decisions.

## 1 Introduction

Decision makers need to understand the consequences of their decisions prior to making them. When decisions are based on predictions from a machine learning model, the decision loss – the loss incurred under a decision rule based on the predictions – summarizes the consequences of these decisions. As an example, suppose a machine learning practitioner develops a model to predict patient length-of-stay in the hospital [17, 3]. A hospital decides whether they have capacity to admit new patients based on the model's predictions of current patients' length-of-stay (e.g. for each current patient who is predicted to have a length-of-stay that is less than $k$ days, the hospital schedules a new patient). Incorrect decisions due to the model's predictions cause the hospital to accrue costs from under-utilizing resources or overbooking procedures. The decision loss is an aggregation of the costs incurred from incorrect decisions. To determine whether a decision rule is safe to use, the hospital would like to have an accurate estimate of the decision loss under different choices of $k$ and different costs associated with errors. This type of decision-making scenario occurs in many high-stakes settings such as designing interventions for adverse weather events [33, 9] and resource allocation decisions using economic estimates [15, 32].

Probabilistic predictions (probabilistic forecasts) can be used to estimate decision loss prior to deployment. In this work, we consider the regression setup, where a forecast is represented by a cumulative distribution function over the possible outcomes. If the forecasted probabilities of

35th Conference on Neural Information Processing Systems (NeurIPS 2021).

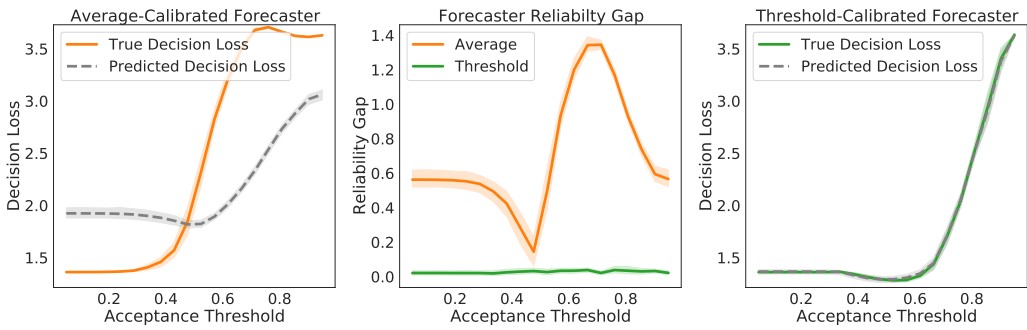

Figure 1: We evaluate average-calibrated and threshold-calibrated patient length-of-stay forecasters across a range of threshold decision rules. **Left**: The average-calibrated forecaster underestimates the true decision loss for some decision rules and overestimates it on others, resulting in a nonzero reliability gap. **Middle**: The reliability gap is minimized under the threshold-calibrated forecaster but not under the average-calibrated forecaster. **Right**: The threshold-calibrated forecaster accurately predicts the true decision loss across a range of decision rules.

incorrect decisions match the true frequencies of these events, then the average decision loss can be accurately predicted from the forecasts. However, forecasted probabilities of incorrect decisions do not typically match the true ones, yielding inaccurate decision loss predictions. We refer to the absolute difference between the average loss predicted by forecaster and the true average loss as the **reliability gap**.

Many previous works in calibration and uncertainty quantification are motivated by the assumption that calibrated uncertainty estimates will yield safer or more reliable downstream decisions [31, 2, 22, 24, 25]. However, we find that the standard notion of calibration, average calibration, does not guarantee zero reliability gap for even a simple class of decision rules: threshold decision rules (left, Figure 1). In a threshold decision, a decision maker takes one of two possible actions depending on whether an outcome falls above or below a cutoff (e.g. the hospital schedules a new patient if a current patient's length-of-stay is less than 3 days, otherwise the hospital does not schedule a new patient). Stronger calibration properties, such as distribution calibration [31], are theoretically guaranteed to yield zero reliability gap but are difficult to achieve in practice. In particular, flexible distribution families can better approximate the true distribution than simple ones and yield lower decision loss, but applying distribution calibration to such forecasters can increase the decision loss and enlarge the reliability gap compared to the uncalibrated forecaster. Thus, existing calibration definitions are either insufficient or impractical for minimizing the reliability gap under threshold decision rules.

To address these shortcomings, we propose a new notion of calibration called **threshold calibration**. Threshold calibration strikes a balance between average and distribution calibration; it is exactly the condition required to guarantee zero reliability gap under threshold decisions and is practical to enforce (Figure 1, Right). First, we establish that threshold calibration is the necessary and sufficient condition to guarantee zero reliability gap for any threshold decision under any threshold loss. Second, we design an *efficient* algorithm that takes an uncalibrated forecaster as input and provably outputs a threshold-calibrated forecaster. Third, we show that empirically, threshold calibration is a *practical* solution; in two real-world settings and a suite of benchmark regression tasks, we find that threshold calibration minimizes the reliability gap across decision makers with different threshold loss functions while achieving similar or improved decision loss compared to the baselines.

## 2 Preliminaries

### 2.1 Notation and Forecasting Setup

We consider the regression setup with a feature space $\mathcal{X}$ and a label space $\mathcal{Y} \subset \mathbb{R}$. The input is a random variable $X \in \mathcal{X}$ and the label is a random variable $Y \in \mathcal{Y}$. We use capital letters to denote random variables $X, Y$ and lower case letters to denote their values $x, y$.

Let $\mathcal{F}(\mathcal{Y})$ be the space of cumulative distribution functions (CDFs) over $\mathcal{Y}$. A forecaster $h : \mathcal{X} \to \mathcal{F}(\mathcal{Y})$ is a function that maps an input from the feature space to a CDF on $\mathcal{Y}$. In other

words, given a fixed input $x \in \mathcal{X}$, the forecaster outputs the predicted CDF $h[x] \in \mathcal{F}(\mathcal{Y})$. Ideally, the forecaster aims to predict the CDF of $Y$ given $X$.

To further clarify the notation, for a fixed input-label pair $(x, y) \in \mathcal{X} \times \mathcal{Y}$, $h[x]$ is a CDF over the predicted label values and $h[x](y) \in [0, 1]$ is the value of the CDF $h[x]$ at the point $y$. We note that $h[X]$ is a random variable that takes values in $\mathcal{F}(\mathcal{Y})$ and $h[X](Y)$ is a random variable that takes values in $[0, 1]$.

Let $h^*[X]$ be the true conditional CDF of $Y$ given $X$. We use $\sim$ to denote the distribution of a random variable. We have that $Y \sim h^*[X]$. We introduce a new random variable $\tilde{Y}$ to represent a label distributed according to the $h[X]$, the forecasted conditional distribution, so $\tilde{Y} \sim h[X]$.

## 2.2 Decision-Making

Let $\mathcal{A}$ be a countable action space. A decision rule $\delta : \mathcal{X} \to \mathcal{A}$ is any map from an input $x$ (e.g. a current patient's attributes) to an action $a$ (e.g. admit a new patient). We assume that a decision maker has a loss function $\ell : \mathcal{X} \times \mathcal{Y} \times \mathcal{A} \to \mathbb{R}$, describing the loss incurred when choosing an action $a$ on an input-label pair $(x, y)$. Because the labels $y$ are unobserved, the decision maker often wants to minimize their expected loss assuming that the labels are distributed according to the forecasted distribution. As a result, they use the Bayes decision rule with respect to $h$.

**Definition 1** (Bayes Decision Rule). *Given a space of decision rules $\Delta$, the Bayes decision rule with respect to the forecaster $h$ is the decision rule in $\Delta$ that minimizes the expected loss under the forecasted distribution*

$$\delta_h^* = \arg \inf_{\delta \in \Delta} \mathbb{E}_X \mathbb{E}_{\tilde{Y} \sim h[X]}[\ell(X, \tilde{Y}, \delta(X))]]$$

## 2.3 Threshold Decisions

We focus on the setting where the decision maker aims to minimize a threshold loss function. The action space $\mathcal{A}$ consists of two actions so $\mathcal{A} = \{0, 1\}$. A threshold loss function $\ell$ is defined as follows

$$\ell(x, y, a) = \sum_{i \in \{0,1\}} c_{1,i} \mathbb{I}(y \leq y_0, a = i) + \sum_{i \in \{0,1\}} c_{0,i} \mathbb{I}(y > y_0, a = i),$$

where $c_{i,j} \in \mathbb{R}$. The $c_{i,j}$'s denote *decision costs*, costs associated with different outcome-action pairs, and $y_0$ is a *decision threshold*. Let $\mathcal{L}$ be the space of threshold loss functions, which are all losses of this form with any $c_{i,j} \in \mathbb{R}$ and $y_0 \in \mathbb{R}$.

Given a threshold loss function $\ell$, the decision maker can use the Bayes decision rule $\delta_h^*$ in Definition 1 to select which action to take. We show that the resulting decision rules always take the form of

$$\delta_h^*(x) = \mathbb{I}(h[x](y_0) \geq \alpha) \text{ or } \delta_h^*(x) = \mathbb{I}(h[x](y_0) \leq \alpha)$$

for some parameters $\alpha \in [0, 1]$ and $y_0 \in \mathcal{Y}$ that depends on the loss function (proved in Appendix B). We call such decision rules **threshold decision rules** because intuitively, they choose the action based on whether $h[x](y_0)$ is greater (or less) than a threshold $\alpha$. We denote the space of such decision rules as $\Delta_h$. Since the decision maker's loss function is a threshold loss function, the decision maker can restrict the space of decision rules they consider to threshold decision rules on the forecasted CDFs.

# 3 Reliable Decision-Making with Threshold Calibration

## 3.1 Problem Setup

Forecasts are often produced by one group, such as machine learning practitioners or scientists, and consumed by another, such as policy makers or private agents [14]. Motivated by this paradigm, we model these two entities separately:

1. A forecaster $h$ takes inputs $x \in \mathcal{X}$ and produces CDFs $h[x]$ over the possible outcomes in $\mathcal{Y}$. The provider of $h$ does not know the specific downstream tasks for which $h$ is used.

2. A decision maker has a dataset of unlabeled inputs $\mathcal{D} = \{x_i\}_{i=1}^n$, binary action space $\mathcal{A} = \{0, 1\}$, and a threshold loss function $\ell \in \mathcal{L}$ of interest. The decision maker must take an action $a_i \in \mathcal{A}$ for each unlabeled input $x_i$. The decision maker uses the forecaster $h$ to select $\{a_i\}_{i=1}^n$ because (1) the decision maker does not have enough labeled data to build their own model locally or (2) building the model requires a domain expert.

Multiple decision makers may rely on the same forecaster but have different loss functions. Further, a decision maker's loss function can change if their decision costs or decision threshold change. In this setting, we identify the conditions on $h$ that the provider can enforce to ensure reliable decision-making under threshold decisions.

## 3.2 Reliability Gap

Decision makers often need to accurately estimate the average decision loss incurred under a decision rule prior to deployment. To quantify the accuracy of these decision loss predictions, we define the reliability gap.

**Definition 2** (Reliability Gap). *Given a forecaster $h$, we define the the reliability gap $\gamma(\delta, \ell)$ of a particular decision rule $\delta$ under a loss function $\ell$ as*

$$\gamma(\delta, \ell) = |\mathbb{E}_X \mathbb{E}_{\tilde{Y} \sim h[X]}[\ell(X, \tilde{Y}, \delta(X))] - \mathbb{E}_X \mathbb{E}_{Y \sim h^*[X]}[\ell(X, Y, \delta(X))]|.$$

The first term in the equation is the average decision loss predicted by the forecaster. Under the forecasted distribution, the labels $\tilde{Y}$ are distributed according to $h[X]$. As a result, the first term does not depend on the true labels and can be computed by the decision maker using the unlabeled data prior to deployment. The second term is the true average decision loss. Under the true conditional distribution, the labels $Y$ are distributed according to $h^*[X]$. So, the second term can be thought of as the loss that is incurred at test-time. One caveat is that the reliability gap quantifies the reliability of *average* decision loss prediction and obtaining zero reliability gap does not imply any instance-based guarantees for individual decisions.

When the forecaster perfectly matches the true distribution (i.e. $h = h^*$), we have $\gamma(\delta, \ell) = 0$ for any decision rule $\delta$ and any loss function $\ell$. However, in practice, we cannot assume that the forecaster predicts the true distribution. In addition, we would like the forecaster to be applicable for different downstream decision makers. As a result, we study the necessary and sufficient conditions on the forecaster that guarantee zero reliability gap for any threshold decision on the forecasted CDFs and any threshold loss function.

## 3.3 Threshold Calibration

We define the property of threshold calibration and show that it is necessary and sufficient to ensure zero reliability gap under any threshold decision on the forecasted CDFs and any threshold loss function. The lemma and theorem in this section are proven in Appendix B.

We define the property of threshold calibration below.

**Definition 3** (Threshold Calibration). *A forecaster $h$ satisfies threshold calibration if*

$$\Pr[h[X](Y) \le c \mid h[X](y_0) \le \alpha] = c \quad \forall y_0 \in \mathcal{Y}, \alpha \in [0, 1], \forall c \in [0, 1]. \tag{1}$$

A threshold-calibrated forecaster is average-calibrated on subsets of the predicted CDFs that satisfy $h[X](y_0) \le \alpha$. We make the following observation about conditioning on the complementary predicted CDFs.

**Lemma 1.** *Given a forecaster $h$ that satisfies Definition 3, then we have that $\forall y_0 \in \mathcal{Y}, \alpha \in [0, 1], \forall c \in [0, 1], \Pr[h[X](Y) \le c \mid h[X](y_0) > \alpha] = c$.*

In a threshold decision task, a decision maker will take action $a$ given inputs with predicted CDFs satisfying $h[X](y_0) \le \alpha$ (and take a complementary action given inputs with predicted CDFs satisfying $h[X](y_0) > \alpha$). Intuitively, threshold calibration ensures that the forecaster satisfies average calibration on the subsets of predicted CDFs where the decision maker chooses $a = 0$ and $a = 1$.

Threshold calibration is a specific type of group calibration [28], where calibration across the collection of groups $\mathcal{G} = \{(X, Y) \in \mathcal{X} \times \mathcal{Y} \mid h[X](y_0) \leq \alpha\}_{y_0 \in \mathcal{Y}, \alpha \in [0,1]}$ is desired. Since threshold calibration requires achieving calibration on intersecting groups, it is also related to the notion of multicalibration [18]. In Section 4, we give an efficient algorithm for achieving threshold calibration that is inspired by previous work on multicalibration.

Using Definition 3 and Lemma 1, we define the threshold calibration error (TCE) to measure deviation from threshold calibration at a threshold $y_0 \in \mathcal{Y}$ and quantile $\alpha \in [0, 1]$.

**Definition 4** (Threshold Calibration Error).

$$TCE(h, y_0, \alpha) = \int_0^1 |\Pr[h[X](Y) \leq c \mid h[X](y_0) \leq \alpha] - c| \quad dc$$

$$+ \int_0^1 |\Pr[h[X](Y) \leq c \mid h[X](y_0) > \alpha] - c| \quad dc.$$

Threshold calibration is a desirable property due to its connection to achieving zero reliability gap.

**Theorem 1.** *Let $\mathcal{L}$ be the space of threshold loss functions. Given a forecaster $h$, let $\Delta_h$ be the space of threshold decision rules on the forecasted CDFs of $h$. A forecaster $h$ satisfies threshold calibration if and only if $\gamma(\delta, \ell) = 0 \quad \forall \delta \in \Delta_h, \forall \ell \in \mathcal{L}$.*

We obtain this result by observing that the expected decision loss under the true distribution can be decomposed into two terms. The first term corresponds to the cost incurred from "false positive" errors and the second term corresponds to the cost incurred from "false negative" errors. Under threshold calibration, the forecaster's predicted error rates match the true error rates. Since the decision loss (with any choice of costs) is a linear combination of these error rates, the expected decision loss predicted by the forecaster matches the expected decision loss under the true distribution. Thus, under a threshold-calibrated forecaster, we achieve zero reliability gap under any threshold decision on the forecasted CDFs and any threshold loss function.

### 3.4 Comparison to Existing Calibration Definitions

We compare threshold calibration to other methods for calibrating probabilistic forecasts. Average calibration is the standard definition of calibration for regression [23, 12].

**Definition 5** (Average Calibration). *A forecaster $h$ satisfies average calibration if*

$$\Pr[h[X](Y) \leq c] = c \quad \forall c \in [0, 1].$$

In other words, a forecaster is average-calibrated if the true label $Y$ is below the $c$-th quantile of the forecasted CDF $h[x]$ exactly $c$ percent of the time.

In contrast, distribution calibration is a much stronger definition of calibration [31]. Intuitively, distribution calibration requires a forecaster to be calibrated for every distribution in the forecaster's model family.

**Definition 6** (Distribution Calibration). *A forecaster $h$ satisfies distribution calibration if*

$$\Pr[h[X](Y) \leq c \mid h[X] = g] = c \quad \forall g \in \mathcal{F}(\mathcal{Y}),$$

*where $\mathcal{F}$ is space of CDFs corresponding to the forecaster's model family.*

We outline the relationship between average, threshold, and distribution calibration in the following proposition.

**Proposition 1.** *If a forecaster satisfies distribution calibration, then it satisfies threshold calibration. If a forecaster satisfies threshold calibration, then it satisfies average calibration.*

We note that the converses of the statements in Proposition 1 are not necessarily true. A threshold-calibrated forecaster does not necessarily satisfy distribution calibration. An average-calibrated forecaster does not necessarily satisfy threshold calibration or distribution calibration (see Appendix C). This implies that an average-calibrated forecaster does not satisfy the necessary condition of Theorem 1, meaning that the reliability gap under threshold decisions may not be zero. So, decision

makers who rely on a forecaster that only satisfies average calibration (but not threshold calibration) are not guaranteed to accurately estimate their decision loss under threshold decisions.

From Proposition 1, we have that a distribution-calibrated forecaster satisfies the necessary condition of Theorem 1. However, distribution calibration can be challenging to achieve in practice because the same CDF is rarely predicted more than one time on the training samples, making it difficult to guarantee calibration without compromising the *sharpness* of the forecasts. Sharpness corresponds to the width of the prediction intervals generated from the forecasts, and sharp forecasts yield short prediction intervals. Although distribution calibration is theoretically guaranteed to yield zero reliability gap, we observe that achieving distribution calibration is challenging when the model family is complex (Section 5).

Finally, we emphasize the threshold calibration is exactly the condition needed to guarantee the reliability gap is zero in Theorem 1.

## 4 Achieving Threshold Calibration

We design a recalibration algorithm that takes an uncalibrated forecaster as input and provably outputs a threshold-calibrated forecaster. Our algorithm is an iterative procedure that terminates when the maximum TCE is less than a user specified threshold $\epsilon$. Our key result is that the algorithm must terminate after $O(1/\epsilon^2)$ iterations.

Pseudo-code for the algorithm is shown in Algorithm 1. Intuitively, at each iteration of the algorithm, we find the $y_0^t$ and $\alpha^t$ where the TCE in Definition 4 is maximized. This partitions the input $\mathcal{X}$ into two parts: those where $h[x](y_0^t) \leq \alpha^t$ and those where $h[x](y_0^t) > \alpha^t$. For each partition, we use a standard recalibration algorithm (Isotonic regression [23]) to achieve average calibration. Intuitively, after the recalibration step, the forecaster should satisfy average calibration for each partition, and hence the TCE in Definition 4 must be (close to) 0 for $y_0^t$ and $\alpha^t$. We repeat this procedure until the TCE is less than $\epsilon$ for every possible $y_0$ and $\alpha$.

---

**Algorithm 1:** Threshold Recalibration

1 **Input**: Forecaster $h : \mathcal{X} \to \mathcal{F}(\mathcal{Y})$, maximum error $\epsilon > 0$
2 **Output**: A threshold-calibrated forecaster
3 Set $h^0 \leftarrow h$
4 **for** $t = 1, 2, \cdots$ *until maximum threshold calibration error* $\sup_{y_0, \alpha} TCE(h^{t-1}, y_0, \alpha) \leq \epsilon$ **do**
5      Find the $y_0$ and $\alpha$ that maximize threshold calibration error.
       $y_0^t, \alpha^t \leftarrow \arg\sup_{(y_0, \alpha) \in \mathcal{Y} \times [0,1]} \text{TCE}(h^{t-1}, y_0, \alpha)$
6      Partition input features $\mathcal{X}$ into $\mathcal{X}_0 \leftarrow \{x \in \mathcal{X} \mid h^{t-1}[x][y_0^t] \leq \alpha^t\}$ and $\mathcal{X}_1 = \mathcal{X} \setminus \mathcal{X}_0$.
7      Use Isotonic regression to learn recalibration maps $\phi_0^t, \phi_1^t : \mathcal{F}(\mathcal{Y}) \to \mathcal{F}(\mathcal{Y})$ on $\mathcal{X}_0$ and $\mathcal{X}_1$
       respectively.
8      Apply the recalibration map to obtain new prediction functions.
       $h^t[x] \leftarrow \begin{cases} \phi_0^t(h^{t-1}[x]) & \text{if } x \in \mathcal{X}_0 \\ \phi_1^t(h^{t-1}[x]) & \text{otherwise} \end{cases}$
9 **end**
10 **return** $h^T$ where $T$ is the final iteration count.

---

The following theorem shows that our iterative threshold calibration procedure converges in a small number of iterations. The intuition of the proof is that after each iteration, the L2 distance between the prediction functions $h$ and the true CDF $h^*$ must decrease by at least $\epsilon^2$. Therefore, the algorithm must terminate before the L2 distance decreases below 0 (which is impossible). A full proof is provided in Appendix B.

**Theorem 2.** *Algorithm 1 converges after at most $O(1/\epsilon^2)$ iterations and outputs a forecaster with threshold calibration error at most $\epsilon$.*

For simplicity, we do not consider finite sample approximation of the TCE in line 5 of Algorithm 1. Line 5 can be interpreted in two ways: line 5 estimates the TCE on the true distribution (which we can only do with infinite samples), or on the empirical distribution (i.e. the uniform distribution on

the recalibration data). Under the former interpretation, Theorem 2 holds assuming that line 5 can estimate the true TCE (which is the ideal scenario with infinite data). Under the latter interpretation, Theorem 2 holds for the empirical distribution, i.e. it guarantees that Algorithm 1 will output a forecaster with threshold calibration error at most $\epsilon$ *on the empirical distribution* rather than the true distribution. We will instead use experiments to show that Algorithm 1 can generalize to the true distribution. Note that under both interpretations, Algorithm 1 will converge after at most $O(1/\epsilon^2)$ iterations. For completeness, we describe the finite sample version of the algorithm in Appendix A.

## 5 Experiments

In the following experiments, we demonstrate that threshold calibration can minimize the reliability gap (1) across a range of decision costs, (2) across a range of decision thresholds, and (3) in simple and complex model families. Across all datasets and forecaster model families that we consider, we find that threshold calibration outperforms the baselines in reducing the size of the reliability gap while attaining similar or improved decision loss compared to the baselines.

### 5.1 Datasets

We consider datasets that relate to real-world decision-making tasks and standard benchmarks. In the main paper, we show results on the UCI Protein and the MIMIC-III datasets. All remaining results can be found in Appendix A.

**MIMIC-III.** Patient length-of-stay predictions are used for hospital scheduling and resource management [17]. We consider a patient length-of-stay forecaster trained on patient admission laboratory values from the MIMIC-III dataset [20]. In our decision task, the hospital decides to schedule a new patient for an elective procedure if a current patient is predicted to have a short length of stay.

**Demographic and Health Survey (DHS).** Local wealth measurements are used to inform resource allocation decisions. We use the DHS data from Sheehan et al. [30] to predict asset wealth from satellite images as done in Yeh et al. [32] and Sheehan et al. [30]. Our experimental setup is motivated by the decision task defined in Yeh et al. [32], where aid is allocated to regions where the predicted asset wealth falls below a particular threshold.

**UCI Regression Datasets.** We evaluate on a suite of UCI regression datasets (Naval, Protein, Energy, Crime) [11]. They are common benchmarks in the uncertainty quantification literature [31, 2, 8, 23].

### 5.2 Experimental Setup and Baselines

**Experimental Setup.** We consider a forecaster that outputs Gaussian distributions and a forecaster that outputs Gaussian-Laplace mixture distributions. We use a train/validation/test split. The uncalibrated forecaster is a neural network trained on the training set with the validation set used for early stopping. For large datasets (Protein, Energy, Naval, MIMIC-III), the recalibration transform is trained on the validation set. For small datasets (Crime, DHS), the recalibration transform is trained on the training and validation set. On the test set, we evaluate our method and the baselines using decision-making metrics (Section 5.3). Calibration metrics are also measured and results are provided in Appendix A.

**Baselines.** We compare the uncalibrated forecaster to the forecaster after enforcing average, threshold, or distribution calibration through a posthoc recalibration procedure. Methods for achieving these properties are described in Appendix A.

### 5.3 Decision-Making Metrics

We simulate decision makers enumerated $i = 1, 2 \ldots M$ who use a probabilistic forecaster $h$ for their threshold decision tasks. We assume that there is no cost associated with true positives or true negatives, and the total cost of a false positive plus a false negative is equal to 10 for all decision makers. As a result, decision maker $i$'s task is determined by a decision threshold $y_0^i$ and decision cost ratio $c_i$. Each decision maker has a loss function $\ell^i(x, y, a) = 10c_i\mathbb{I}(a = 1, y \geq y_0^i) + 10(1 - c_i)\mathbb{I}(a = 0, y < y_0^i)$ and a decision rule

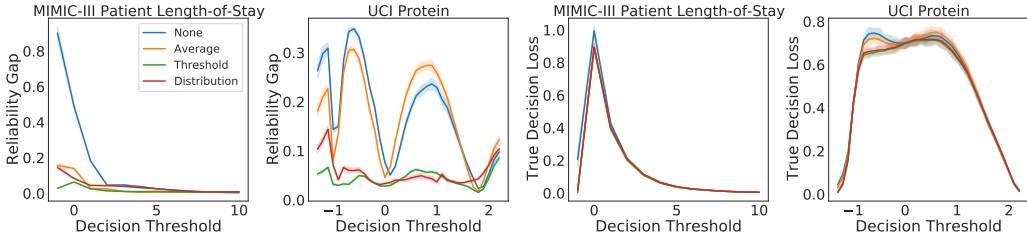

Figure 2: Under the Gaussian forecaster and across different decision thresholds, threshold calibration reduces the reliability gap on both datasets while average calibration does not reduce the reliability gap on the Protein dataset (**Left, Middle Left**), and all calibration methods yield improved or comparable decision loss compared to the uncalibrated forecaster (**Middle Right, Right**). Error bars represent 95% confidence intervals and are generated over 6 random trials.

$\delta_{h,\alpha}^i(x) = \mathbf{1}(h[x](y_0^i) \geq \alpha)$. We consider decision makers with $(y_0^i, c_i) \in \mathcal{Y}_0 \times \mathcal{C}$ where $\mathcal{Y}_0$ and $\mathcal{C}$ each consist of 50 uniformly spaced points that span the label space and [0.05, 0.95], respectively.

For each decision maker $i$, we compute the decision loss (the loss incurred by the Bayes decision rule $\delta_h^{*,i}(X)$) and the reliability gap (averaged over the possible threshold decision rules).

$$\text{Decision Loss} = \mathbb{E}_X \mathbb{E}_{Y \sim h^*[X]}[\ell^i(X, Y, \delta_h^{*,i}(X))] \quad \text{Reliability Gap} = \frac{1}{|\mathcal{C}|} \sum_{\alpha \in \mathcal{C}} |\gamma(\delta_{h,\alpha}^i, \ell^i)|.$$

Aggregate statistics can be obtained by averaging over all $M$ decision makers, all decision makers who share the same threshold $y_0$, or all decision makers who share the same cost ratio $c$.

## 5.4 Results

Using the MIMIC-III and UCI Protein datasets, we study the effect of recalibration on the reliability gap and the decision loss achieved by decision makers with different decision thresholds and cost ratios. Furthermore, we examine the effect of recalibration on forecasters that output CDFs from simple (Gaussian) and complex (Gaussian-Laplace mixture) model families.

**Threshold Calibration Minimizes Reliability Gap Across Decision Thresholds.** We evaluate the effect of recalibrating the Gaussian forecaster on decision makers with different decision thresholds. On both datasets, threshold calibration yields the largest decrease in the reliability gap (left plots, Figure 2). Distribution calibration also decreases the reliability gap across decision thresholds, relative to the uncalibrated Gaussian forecaster. Average calibration does not consistently reduce the reliability gap; on the UCI Protein dataset, the reliability gap of the average-calibrated forecaster enjoys a slight decrease at some decision thresholds but is increased at others, relative to the uncalibrated Gaussian forecaster (middle left, Figure 2). Lastly, these calibration methods achieve similar decision loss to the uncalibrated forecaster (right plots, Figure 2). These trends are consistent with the results obtained on the other datasets under the Gaussian forecaster. Threshold calibration outperforms baselines across different decision thresholds under the Gaussian-Laplace forecaster, as well (Appendix A).

**Threshold Calibration Minimizes Reliability Gap Across Decision Cost Ratios.** Across decision makers with different cost ratios, distribution and threshold calibration reduce the reliability gap relative to the uncalibrated forecaster, with threshold calibration yielding the largest decreases in the reliability gap (left plots, Figure 3). Meanwhile, average calibration does not consistently reduce the reliability gap; on the UCI Protein dataset, it achieves similar reliability gap to the uncalibrated forecaster (middle left, Figure 3). As before, these calibration methods achieve similar decision loss to the uncalibrated forecaster (right plots, Figure 3). These trends are consistent with results obtained on the other datasets under the Gaussian forecaster. Threshold calibration outperforms baselines across different decision cost ratios under the Gaussian-Laplace forecaster, as well (Figure 4).

**Distribution Calibration Degrades Performance under Complex Model Families.** Forecasters that can output CDFs from more flexible model families (e.g. Gaussian-Laplace mixture distributions)

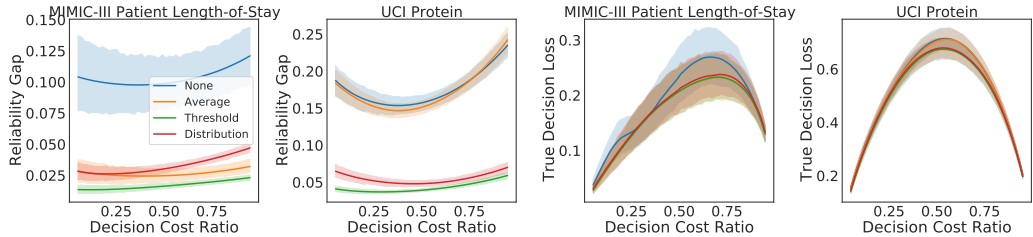

Figure 3: Under the Gaussian forecaster and across different decision cost ratios, threshold calibration reduces the reliability gap on both datasets while average calibration does not reduce the reliability gap on the Protein dataset (**Left, Middle Left**), and all calibration methods yield improved or comparable decision loss compared to the uncalibrated forecaster (**Middle Right, Right**). Error bars represent 95% confidence intervals and are generated over 6 random trials.

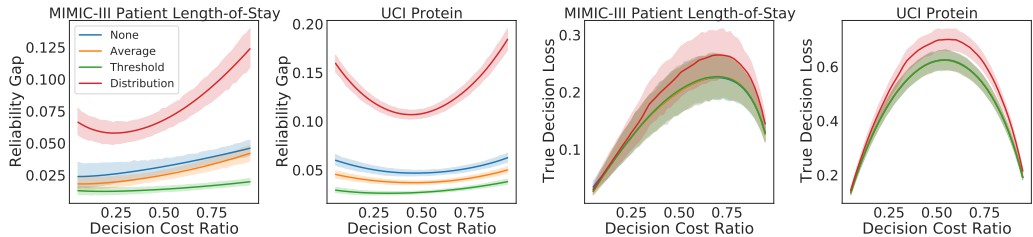

Figure 4: We consider the effect of recalibrating the Gaussian-Laplace forecaster under a range of decision cost ratios. Threshold calibration reduces the reliability gap while distribution calibration can enlarge the reliability gap (**Left, Middle Left**). Average and threshold calibration achieve comparable or lower decision loss as the baseline forecaster, while distribution calibration increases the decision loss. Error bars represent 95% confidence intervals and are generated over 6 random trials.

may be able to better capture the true conditional distribution of $Y$ given $x$ compared to Gaussian forecasters. As a result, we examine the effect of the recalibration procedures when the uncalibrated forecasts follow a more flexible distribution. The uncalibrated Gaussian-Laplace forecaster (Figure 4) yields a smaller reliability gap and smaller decision loss compared to the uncalibrated Gaussian forecaster (Figure 3). Applying threshold calibration to the Gaussian-Laplace forecaster further reduces the reliability gap. However, under the Gaussian-Laplace forecaster, distribution calibration enlarges the size of the reliability gap and increases the decision loss. Although distribution calibration is theoretically guaranteed to minimize the reliability gap, it is challenging to achieve in finite samples without compromising the *sharpness* of the forecasts (in our case, decision loss). So, we find that decision loss and reliability gap increase. We hypothesize that the recalibration dataset may not contain many instances that yield similar distribution parameters, so the recalibration transform does not generalize well to unseen data. We also observe these trends the UCI Crime, UCI Energy, and DHS datasets.

## 6 Related Work

**Forecasting and Decision Making.** The connection between forecasts and decision making was first studied in economics [1, 29]. The development of Bayesian decision analysis connected topics of forecasts and decision-based loss functions [10, 4]. Decision-making under uncertainty with probabilistic forecasts was then studied in econometrics [7]. [19] also considers learning regression functions that minimize a decision loss. While [19] focuses on transforming the predicted CDF to a point prediction, our method focuses on transforming the predicted CDF into a new CDF. [19] also requires knowing the loss function to learn the transformation, while our method assumes that the loss function belongs to a commonly used function family (threshold loss functions).

**Calibration.** Calibration definitions have been studied in the statistics literature [5, 6, 26]. For the regression setting, methods for ensuring that machine learning models satisfy average calibration

have been studied in [23, 8]. In addition, methods for achieving stronger calibration notions have also been introduced such as distribution calibration [31] and individual calibration [34]. Calibration and trustworthy predictions in the medical domain are also studied in [16]. [16] introduces the notion of D-calibration, which is related to our average calibration baseline, but is tailored to the survival analysis task. A perfectly average calibrated prediction function is also D-calibrated, and vice versa.

**Multicalibration.** Our definition of threshold calibration is most related to the line of work on multicalibration [18, 21]. Given a large collection $\mathcal{G}$ of potentially intersecting groups of the data, a predictor is multicalibrated on $\mathcal{G}$ if it is simultaneously calibrated on every sufficiently large group in $\mathcal{G}$ [18]. Previous works give methods for achieving mean and moment multicalibration for predictor functions. Our iterative procedure for achieving threshold calibration is inspired by methods for achieving multicalibration.

## 7   Limitations and Societal Impact

Our work demonstrates that certain types of calibration enable decision makers to estimate decision loss before deployment, which should not be confused with enabling decision makers to make optimal decisions. For example, a forecaster that always outputs the marginal distribution of $Y$ is threshold-calibrated but likely incurs high decision loss. Furthermore, posthoc recalibration is limited by the quality of the baseline model. If the baseline model outputs the marginal distribution of $Y$, then it is already threshold-calibrated but likely is not useful for decision making. Applying our threshold calibration method will not offer any benefit in this case.

Also, our work assumes that predictions of $Y$ do not affect the true label $Y$. However, when predictions are used to make decisions, they can often influence the outcome they aim to predict [27]. Our work does not account for these performative effects, so the decision loss may not be accurately estimated in these settings. Future work could focus on developing calibration procedures that enable forecasters to be robust to such distribution shifts. In addition, we specifically focus on binary-action threshold decisions. Future work may generalize our results to the setting where decision makers have loss functions involving multiple thresholds and multiple actions.

There is a potential for negative societal impact if threshold calibration is incompatible with fairness criteria. Nevertheless, we note that the perfect predictor (that predicts the true conditional probability) satisfies our calibration definition. Consequently, if the perfect predictor satisfies some fairness notion (such as group calibration), then our calibration definition is also compatible with that fairness notion. Note that the perfect predictor does not satisfy a fairness notion called demographic parity, hence our calibration definition is not compatible with demographic parity either.

## 8   Conclusion

We show that a threshold-calibrated forecaster theoretically guarantees accurate decision loss estimation under threshold decision losses and threshold decision rules. We provide an iterative procedure for achieving threshold calibration and show that in practice it minimizes the reliability gap relative to baselines without compromising the forecaster's decision loss. Such estimates permit decision makers to reason about the consequences of their decisions prior to deployment.

## Acknowledgements

RS is supported in part by a NSF GRFP under grant number DGE-1656518. SZ is supported in part by a JP Morgan fellowship and a Qualcomm innovation fellowship. SE is supported in part by NSF(#1651565, #1522054, #1733686), ONR (N000141912145), AFOSR (FA95501910024), ARO (W911NF-21-1-0125) and Sloan Fellowship. We are grateful for Rishi Bommasani, Kristy Choi, Matthew Jörke, Judy Shen, Rui Shu, Fan-Yun Sun, Rohan Taori, Ke Alexander Wang, Rose Wang, and Henry Zhu for insightful discussions.

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
