# A Experimental Setup and Additional Results

## A.1 Reproducibility

We provide a link to our code below. The code includes scripts for downloading the UCI regression datasets. Accessing the MIMIC-III dataset requires an ethics training course and permissions [20], so we do not provide the dataset or download information in the code.

- https://drive.google.com/file/d/12Qh1AWsJcx6UzrRAVYPAYPNemRBj7500/view?usp=sharing.

## A.2 Baseline Forecasters

In our experiments, a forecaster is trained on data $\{(x_i, y_i)\}_{i=1}^n$. We assume the labels $y_i$ are drawn i.i.d. from a distribution with a parameters $\theta_i$. We consider two types of predictive distributions, a unimodal Gaussian distribution and a mixture of a Gaussian and a Laplace distribution. For a Gaussian distribution, the distribution parameters $\theta_i$ consists of a mean $\mu_i$ and standard deviation $\sigma_i$. For a mixture of a Laplace and Gaussian distribution, the distribution parameters $\theta_i$ consist of the weight assigned to the Gaussian component $w_i$, the Gaussian mean $\mu_i$ and standard deviation $\sigma_i$, and the Laplace location $m_i$ and scale $b_i$.

The forecaster is a neural network $h_w$ where $w$ denotes the parameters of the network. The network takes $x$ as input. For the Gaussian forecaster, the network outputs the parameters of a Gaussian distribution (2 parameters). For the Gaussian-Laplace forecaster, the network takes $x$ as input and outputs the parameters of a Gaussian-Laplace distribution (5 parameters). The network can be trained by using negative log likelihood of the predictive distribution as the loss function.

## A.3 Recalibration Procedure

The posthoc recalibration transforms are fit using a recalibration dataset. We detail the recalibration procedure for each type of calibration.

### A.3.1 Average.

To enforce average calibration, we use the method defined in [23], using the recalibration dataset to fit a single isotonic regression with linear interpolation.

### A.3.2 Threshold

We use a finite sample version of the method described in Section 4 to enforce threshold calibration. We run the algorithm for $T = 40$ iterations for all datasets. We give a detailed outline of the algorithm we use.

---

**Algorithm 2:** Threshold Recalibration

---

1 **Input**: Uncalibrated forecaster $h : \mathcal{X} \to \mathcal{F}(\mathcal{Y})$, recalibration dataset $\mathcal{D} = \{x_i, y_i\}_{i=1}^n$, discretization parameter $K \in \mathbb{Z}_+$, number of iterations $T$
2 **Output**: A threshold-calibrated model $h^T : \mathcal{X} \to \mathcal{F}(\mathcal{Y})$.
3 $m, M \leftarrow \inf_{i \in [n]} y_i, \sup_{i \in [n]} y_i$
4 $\mathcal{Y} \leftarrow \{m + \frac{(M-m)j}{K} \mid j = 1, 2, \ldots K\}$
5 $\mathcal{Q} \leftarrow \{\frac{j}{K} \mid j = 1, 2, \ldots K\}$
6 $h^0 \leftarrow h$
7 **for** $t = 1, 2, 3, \ldots T$ **do**
8     Select the $(y_0^t, \alpha^t,)$ that yields the highest TCE.
    $(y_0^t, \alpha^t) \leftarrow \arg\sup_{(y_0, \alpha) \in \mathcal{Y} \times \mathcal{Q}} \widehat{\text{TCE}}(h^{t-1}, y_0, \alpha)$
9     Compute $h^t$ applying Algorithm 3 with the arguments $h^{t-1}, y_0^t, \alpha^t, \mathcal{D}$
10 **end**
11 **return** $h^T$

---

---

**Algorithm 3:** Recalibration at Single Threshold-Quantile Pair

---

1 **Input**: Uncalibrated forecaster $h : \mathcal{X} \to \mathcal{F}(\mathcal{Y})$, recalibration dataset $\mathcal{D} = \{x_i, y_i\}_{i=1}^{n}$, threshold $y_0 \in \mathbb{R}$, quantile $\alpha \in [0, 1]$, discretization parameter $K \in \mathbb{Z}_+$.

2 **Output**: A model $h : \mathcal{X} \to \mathcal{F}(\mathcal{Y})$ that is threshold-calibrated at a threshold $y_0$ and quantile $\alpha$.

3 Partition the data based on whether $h[x_i](y_0) \leq \alpha$:

4 $\mathcal{I}_1 \leftarrow \{i \in [n] \mid h[x_i](y_0) \leq \alpha\}$.

5 $\mathcal{I}_2 \leftarrow [n] \setminus \mathcal{I}_1$.

6 Learn recalibration functions $\mathcal{R}_k$ for each $\mathcal{I}_k$ :

7 **for** $k = 1, 2$ **do**

8      Create recalibration dataset $\mathcal{D}_k \leftarrow \{h[x_i](y_i), \hat{P}_k(h[x_i](y_i))\}_{i \in \mathcal{I}_k}$, where
         $\hat{P}_k(p) \leftarrow |\{i \in \mathcal{I}_k \mid h[x_i](y_i) \leq p\}|/|\mathcal{I}_k|$.

9      Train a model $R_k$ on $\mathcal{D}_k$ (e.g. isotonic regression).

10 $h[x] \leftarrow \begin{cases} R_1(h[x]) & \text{if } h[x](y_0) \leq \alpha \\ R_2(h[x]) & \text{otherwise} \end{cases}$

11 **return** $h$

---

**Recalibration at a Single Threshold-Quantile Pair (Algorithm 3).** Given an uncalibrated model $h$, a recalibration dataset $\mathcal{D}$, and a discretization parameter $K$, we propose a simple recalibration procedure for enforcing calibration for a single threshold-quantile pair $y_0, \alpha$ (Algorithm 3). We give an overview of the algorithm. First, we partition the recalibration samples into two bins, $\mathcal{I}_1$ and $\mathcal{I}_2$, based on whether the predicted CDF value $h[x](y_0)$ is greater than $\alpha$. Next, we learn a recalibration transform $R_k$ for $\mathcal{I}_k$ using a method similar to [23]. To ensure that the recalibrated forecaster outputs valid CDFs, we require that each $R_k : [0, 1] \to [0, 1]$ and is monotonically increasing. For a particular sample $(x, y)$, the appropriate recalibration transform $R_k$ to apply depends on whether $h[x](y_0)$ is greater than $\alpha$.

### A.3.3 Distribution.

To enforce distribution calibration, we construct $p$-dimensional grid where $p$ is the number of parameters in the forecaster's model family. For Gaussian distributions, we have $p = 2$. For Gaussian-Laplace mixture distributions, we have $p = 5$. We set the grid boundaries by computing the range of each distribution parameter on the validation set. We uniformly partition each axis of the grid into $K$ bins. Each validation sample is sorted into a single grid cell based on the predicted distribution parameters. We fit an isotonic regression model (with linear interpolation) as in [23] using the validation samples that fall into a particular grid cell. For evaluation, we sort the test examples into the appropriate grid cell and apply the corresponding recalibration model. We set the number of bins for each parameter to $K = 20$.

### A.4 Calibration Metrics

The expected calibration error (ECE) is used to measure deviations from average calibration [23]. It is defined as follows

$$\text{ECE}(h) = \int_{c \in [0,1]} |\Pr[h[X](Y) \leq c] - c| \quad dc.$$

We contrast this definition with TCE, threshold calibration error, which measures deviations from threshold calibration. Smaller ECE implies better average calibration. Smaller TCE implies better threshold calibration.

### A.5 Hospital Scheduling Decisions on MIMIC-III

### A.5.1 Dataset

Medical Information Mart for Intensive Care III (MIMIC-III) is a freely accessible medical database of critically ill patients admitted to the intensive care unit (ICU) at Beth Israel Deaconess Medical Center (BIDMC) from 2001 to 2012 [20, 13]. During that time, BIDMC switched clinical information

systems from Carevue (2001-2008) to Metavision (2008-2012). To ensure data consistency, only data archived via the Metavision system was used in the dataset.

**Feature Selection.** We select the same patient features and imputed values as in [17]. A total of 17 variables were extracted from the chartevents table to include in the dataset - capillary refill, blood pressure (systolic, diastolic, and mean), fraction of inspired oxygen, Glasgow Coma Score (eye opening response, motor response, verbal response, and total score), serum glucose, heart rate, respiratory rate, oxygen saturation, respiratory rate, temperature, weight, and arterial pH. For each unique ICU stay, values were extracted for the first 24 hours upon admission to the ICU and averaged. Normal values were imputed for missing variables as shown in Table 1. There are 26089 unique ICU stays in the dataset. The final dataset consisted of the total length of ICU stay and the mean value for each of the 17 variables across the first 24 hours.

| Variable | MIMIC-III item ids from chartevents table | Imputed value |
|---|---|---|
| Capillary refll rate | (223951, 224308) | 0 |
| Diastolic blood pressure | (220051, 227242, 224643, 220180, 225310) | 59.0 |
| Systolic blood pressure | (220050, 224167, 227243, 220179, 225309) | 118.0 |
| Mean blood pressure | (220052, 220181, 225312) | 77.0 |
| Fraction inspired oxygen | (223835) | 0.21 |
| GCS eye opening | (220739) | 4 |
| GCS motor response | (223901) | 6 |
| GCS verbal response | (223900) | 5 |
| GCS total | (220739 + 223901 + 223900) | 15 |
| Glucose | (228388, 225664, 220621, 226537) | 128.0 |
| Heart Rate | (220045) | 86 |
| Height | (226707, 226730) | 170.0 |
| Oxygen saturation | (220227, 220277, 228232) | 98.0 |
| Respiratory rate | (220210, 224688, 224689, 224690) | 19 |
| Temperature | (223761, 223762) | 97.88 |
| Weight | (224639, 226512, 226531) | 178.6 |
| pH | (223830) | 7.4 |

Table 1: Variables included in dataset

**Dataset Splits.** For each of 6 random seeds $[0, 1, 2, 3, 4, 5]$, we generate different dataset splits. Given a random seed, we randomly split off 30% of the original dataset to use as the test set. The remaining dataset are further partitioned into a validation set and training set. The validation set consists of 10% of the remaining data.

### A.5.2 Toy Example.

**Setup.** The decision task of interest in Figure 1 is identifying patients with LOS longer than 3.5 days. The optimal decision rule is $\phi(y) = \mathbb{I}(y \geq y_0)$. In the absence of true LOS values, the forecast-based decision rule $\delta(X) = \mathbb{I}(h[x](y_0) \leq \alpha)$ where $\alpha \in [0, 1]$. To evaluate decision rules, we consider a loss function of the form $\ell(x, y, a) = 5\mathbb{I}(a = 1, y < y_0) + 5\mathbb{I}(a = 0, y \geq y_0)$. So, false positives and false negatives incur equal cost and right decisions incur no cost.

**Forecaster Training Procedure.** For each dataset split, we train an average-calibrated forecaster and a threshold-calibrated forecaster.

To obtain a forecaster that obtains perfect average calibration, we train a neural network with 3 hidden layers of 100 units and ReLU activation on training split of the dataset with ECE as the loss function. We check the validation ECE at each epoch and save the model that obtains lowest validation ECE.

To obtain a threshold-calibrated forecaster, we train a neural network with 3 hidden layers of 100 units and ReLU activation on the training split of the dataset with TCE as the loss function. We use a discretization parameter of $K = 100$ for training the forecaster. We check the validation TCE at each epoch and save the model that obtains lowest validation TCE.

We train both forecasters for 300 iterations.

**Results.** Since the forecasters are only trained with a calibration objective, we do not expect them to provide accurate decisions. Nevertheless, we assess the *reliability* of the forecasters by evaluating how

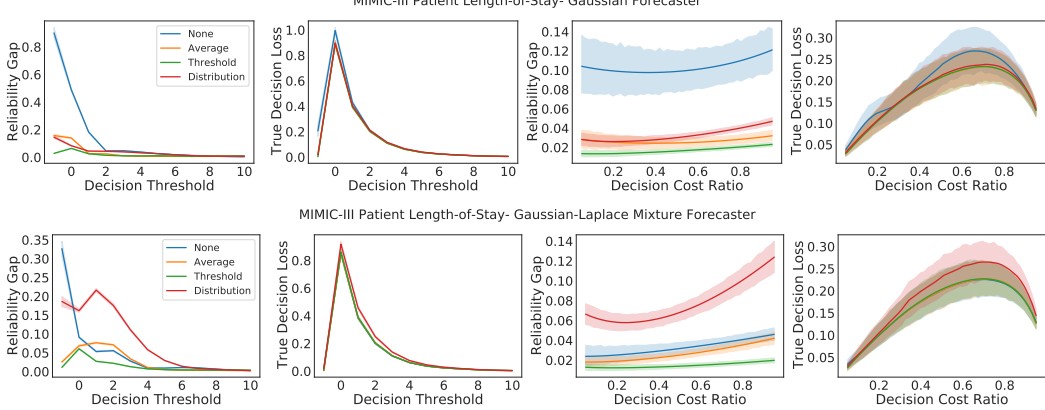

Figure 5: Hospital Scheduling Decisions with Patient Length-of-Stay Forecaster. We plot the decision loss and the reliability gap achieved across decision costs and decision thresholds with various recalibration methods. Threshold calibration achieves the smallest reliability gap among baselines across different decision thresholds and decision cost ratios without compromising the decision loss. Error bars denote 95 % confidence intervals computed over 6 random trials.

well they can predict the true decision loss. The average-calibrated forecaster appears to accurately predict the decision loss at $\alpha = 0.5$, but underestimates the loss at $\alpha > 0.5$ and overestimates the loss at $\alpha < 0.5$.

| Calibration Method | Reliability Gap | TCE | ECE |
|---|---|---|---|
| Average-Calibrated Forecaster | $0.700 \pm 0.342$ | $0.176 \pm 0.013$ | $0.006 \pm 0.002$ |
| Threshold-Calibrated Forecaster | $0.027 \pm 0.020$ | $0.038 \pm 0.008$ | $0.001 \pm 0.002$ |

Table 2: Patient Length-of-Stay Forecasters. The average-calibrated forecaster has a large reliability gap despite achieving near perfect ECE.

### A.5.3 Recalibration Experiment Details.

**Baseline Models.** We train a neural network with 3 hidden layers of 100 units with ReLU activation. The number of inputs to the network is the dimension of the features and the number of outputs of network is the number of parameters of the outputted distribution. This is 2 parameters in the case of outputting Gaussian distributions and 5 parameters in the case of outputting a mixture of a Gaussian and Laplace distribution.

**Baseline Training Procedure.** The baseline models are trained for a maximum of 100 epochs with batch size equal to 128 and we use the Adam optimizer with learning rate 1e-3. Each epoch we check the loss obtained on the validation set and select the model that minimizes the loss on the validation set.

**Recalibration Procedure.** We use the validation set to learn the recalibration transform.

### A.5.4 Recalibration Results.

In Table 3, we show the mean reliability gap and mean decision loss obtained over all $M$ decision makers and TCE and ECE of the forecaster on the MIMIC-III dataset. The standard deviation is computed over 6 randomized trials. The findings are consistent with the findings reported in Section 5; we observe that threshold calibration minimizes the reliability gap and obtains the lowest TCE among the baselines without compromising the decision loss. Although average calibration achieves the lowest ECE across baselines, it does not consistently improve the reliability gap across different decision thresholds. In addition, distribution calibration can increase the reliability gap and decision loss when the model family of the forecaster is more complex (more flexible).

| Forecaster | Method | Reliability Gap | Decision Loss | TCE | ECE |
|---|---|---|---|---|---|
| Gaussian | None | $0.103 \pm 0.019$ | $0.189 \pm 0.008$ | $0.227 \pm 0.017$ | $0.115 \pm 0.01$ |
| | Average | $0.027 \pm 0.005$ | $0.17 \pm 0.005$ | $0.093 \pm 0.019$ | $\mathbf{0.009 \pm 0.004}$ |
| | Threshold | $\mathbf{0.017 \pm 0.006}$ | $\mathbf{0.17 \pm 0.005}$ | $\mathbf{0.055 \pm 0.013}$ | $0.011 \pm 0.003$ |
| | Distribution | $0.033 \pm 0.006$ | $0.173 \pm 0.006$ | $0.061 \pm 0.008$ | $0.011 \pm 0.004$ |
| Gaussian-Laplace | None | $0.033 \pm 0.002$ | $\mathbf{0.165 \pm 0.003}$ | $0.082 \pm 0.01$ | $0.034 \pm 0.003$ |
| | Average | $0.027 \pm 0.005$ | $\mathbf{0.165 \pm 0.003}$ | $0.052 \pm 0.007$ | $\mathbf{0.009 \pm 0.003}$ |
| | Threshold | $\mathbf{0.015 \pm 0.005}$ | $0.166 \pm 0.003$ | $\mathbf{0.048 \pm 0.014}$ | $0.011 \pm 0.002$ |
| | Distribution | $0.077 \pm 0.013$ | $0.188 \pm 0.006$ | $0.115 \pm 0.02$ | $0.032 \pm 0.009$ |

Table 3: Recalibration results for Patient Length-of-Stay Forecasting on MIMIC-III dataset. We observe that threshold calibration procedure decreases the reliability gap.

## A.6 Resource Allocation Decisions on Demographic and Health Survey

### A.6.1 Dataset

We use the satellite images and asset wealth data for African countries of Tanzania, Malawi, Mozambique, Uganda, Rwanda, Zimbabwe from the Demographic and Health Surveys (DHS) from 2009-2011 [30]. We use the nightlight bands of the satellite images. The dataset contains 4191 samples.

**Dataset Splits.** For each of 6 random seeds $[0, 1, 2, 3, 4, 5]$, we generate different dataset splits. Given a random seed, we randomly split off 30% of the original dataset to use as the test set. The remaining dataset are further partitioned into a validation set and training set. The validation set consists of 10% of the remaining data.

### A.6.2 Recalibration Experiment Details

**Baseline Models.** The neural network model that we use is a pretrained Resnet18 architecture from the Pytorch model zoo, which is adjusted to have grayscale inputs. The input shape $255 \times 255 \times 1$ and the number of outputs of network is the number of parameters of the predicted distribution.

**Baseline Training Procedure.** The baseline models are trained for a maximum of 100 epochs with batch size equal to 32 and we use the Adam optimizer with learning rate 1e-3. Each epoch we check the loss obtained on the validation set and select the model that minimizes the loss on the validation set.

**Recalibration Procedure.** Due to the small size of the dataset, the training set and validation set are used to train the recalibration transform.

### A.6.3 Recalibration Results.

In Table 4, we show the mean reliability gap and mean decision loss obtained over all $M$ decision makers and TCE and ECE of the forecaster on the DHS Asset Wealth dataset. The standard deviation is computed over 6 randomized trials. The findings are consistent with the findings reported in Section 5; we observe that threshold calibration minimizes the reliability gap and obtains the lowest TCE among the baselines without compromising the decision loss.

## A.7 UCI Regression Datasets

### A.7.1 Datasets

We use 4 UCI regression datasets. Three of the datasets, Protein, Energy, and Naval, are large and contain 45730, 19735, and 11934 samples respectively. The smaller dataset, Crime, contains 1994 samples.

**Dataset Splits.** For each of 6 random seeds $[0, 1, 2, 3, 4, 5]$, we generate different dataset splits. Given a random seed, we randomly split off 30% of the original dataset to use as the test set. The remaining dataset are further partitioned into a validation set and training set. The validation set consists of 10% of the remaining data.

| Forecaster | Method | Reliability Gap | Decision Loss | TCE | ECE |
|---|---|---|---|---|---|
| Gaussian | None | $0.086 \pm 0.017$ | $0.294 \pm 0.009$ | $0.095 \pm 0.026$ | $0.047 \pm 0.016$ |
| | Average | $0.057 \pm 0.007$ | $0.291 \pm 0.011$ | $0.068 \pm 0.015$ | $\mathbf{0.014 \pm 0.005}$ |
| | Threshold | $\mathbf{0.033 \pm 0.007}$ | $\mathbf{0.287 \pm 0.007}$ | $\mathbf{0.048 \pm 0.013}$ | $0.014 \pm 0.006$ |
| | Distribution | $0.045 \pm 0.019$ | $0.295 \pm 0.009$ | $0.064 \pm 0.027$ | $0.015 \pm 0.008$ |
| Gaussian-Laplace | None | $0.053 \pm 0.015$ | $0.288 \pm 0.003$ | $0.07 \pm 0.022$ | $0.03 \pm 0.015$ |
| | Average | $0.037 \pm 0.01$ | $0.287 \pm 0.004$ | $0.052 \pm 0.021$ | $\mathbf{0.013 \pm 0.005}$ |
| | Threshold | $\mathbf{0.032 \pm 0.005}$ | $\mathbf{0.286 \pm 0.004}$ | $\mathbf{0.045 \pm 0.01}$ | $\mathbf{0.013 \pm 0.005}$ |
| | Distribution | $0.062 \pm 0.016$ | $0.305 \pm 0.004$ | $0.084 \pm 0.028$ | $0.021 \pm 0.014$ |

Table 4: Recalibration results for DHS Survey. We observe that threshold calibration procedure improves the reliability gap.

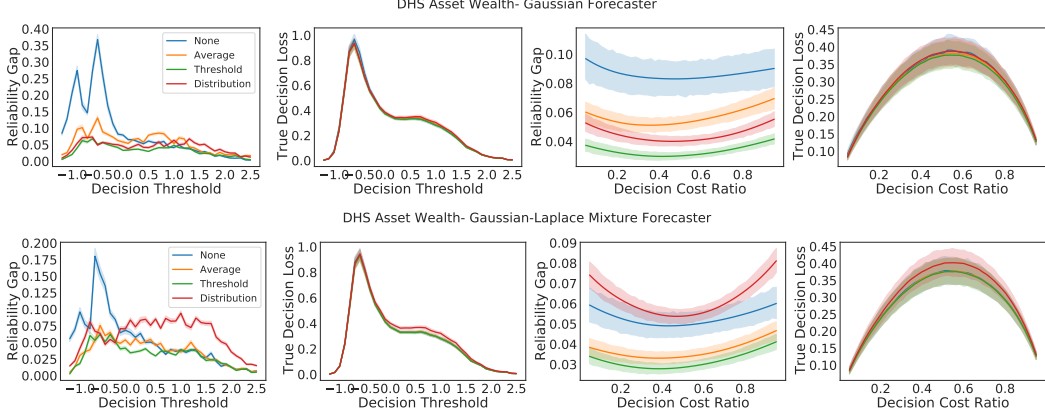

Figure 6: Resource allocation decisions on the DHS Asset Wealth dataset. We plot the decision loss and the reliability gap achieved across decision costs and decision thresholds with various recalibration methods. Threshold calibration achieves the smallest reliability gap among baselines across different decision thresholds and decision cost ratios without compromising the decision loss. Error bars denote 95 % confidence intervals computed over 6 random trials.

#### A.7.2 Recalibration Experiment Details.

**Baseline Models.** We train a neural network with 3 hidden layers of 100 units with ReLU activation. The number of inputs to the network is the dimension of the features and the number of outputs of network is the number of parameters of the outputted distribution. This is 2 parameters in the case of outputting Gaussian distributions and 5 parameters in the case of outputting a mixture of a Gaussian and Laplace distribution.

**Baseline Training Procedure.** The baseline models are trained for a maximum of 100 epochs with batch size equal to 128 and we use the Adam optimizer with learning rate 1e-3. Each epoch we check the loss obtained on the validation set and select the model that minimizes the loss on the validation set.

**Recalibration Procedure.** For the larger datasets (Protein, Naval, Energy), we use the validation set for recalibration to avoid overfitting. For the small dataset (Crime), we combine the training and validation set for recalibration.

#### A.7.3 Recalibration Results.

We show two sets of results. The first includes results with the forecaster that outputs Gaussian distributions in Table 5. The second includes results with the forecaster that outputs Gaussian-Laplace mixture distributions in Table 6. As described in Section 5, we observe that using a forecaster with a more flexible model family (Gaussian-Laplace) can decrease the decision loss and reliability gap of the uncalibrated model. Threshold calibration can further decrease the reliability gap of these models,

while distribution calibration is challenging to achieve and may increase the reliability gap and the decision loss.

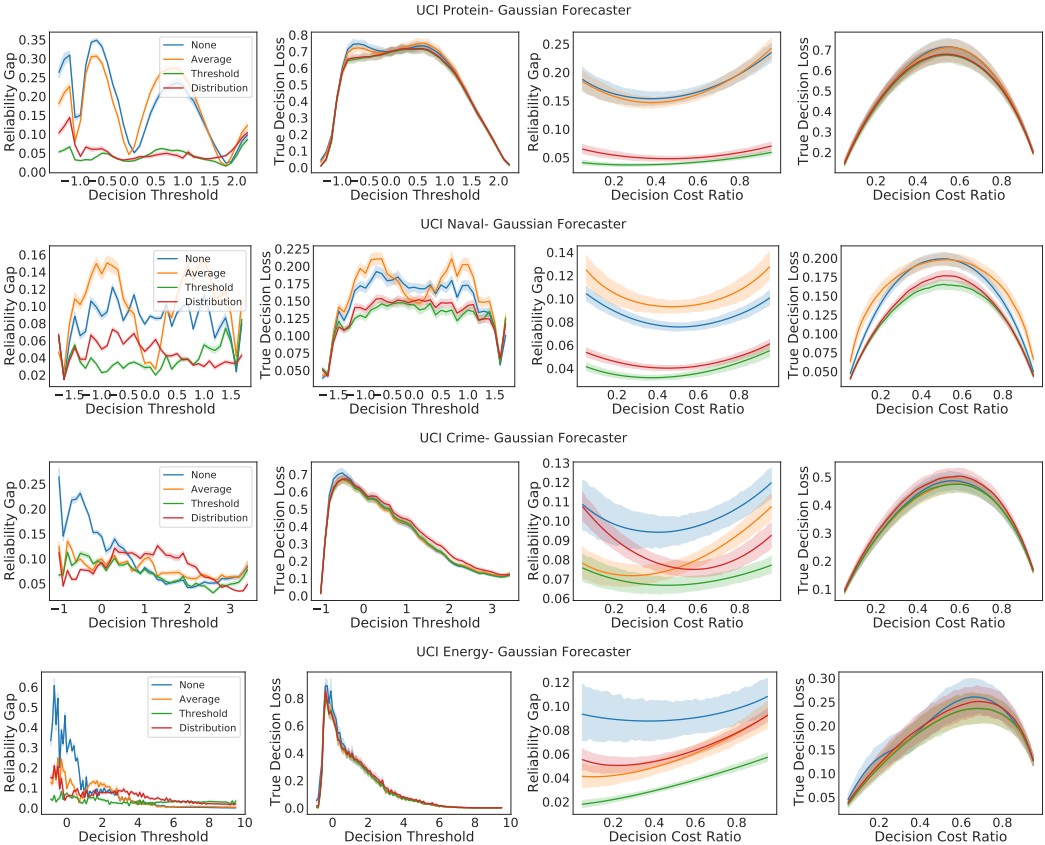

Figure 7: UCI Protein, Naval, Crime, Energy results with Gaussian forecaster. We plot the decision loss and the reliability gap achieved across decision costs and decision thresholds with various recalibration methods. Threshold calibration results in consistent improvements across decision thresholds and decision costs compared to other baselines under the Gaussian forecaster.

# B    Proofs

## B.1    Justification for Threshold Decision Rules

We justify the assumption (from Section 2.3) that a decision maker with a threshold loss function and a decision rule on the forecasted CDFs will restrict the space of decision rules they consider to threshold decision rules on the forecasted CDFs.

Suppose a decision-maker has a threshold loss function and a binary action space $\mathcal{A}$.

$$\ell(x, y, a) = \sum_{i \in \mathcal{A}} \mathbb{I}(y \geq y_0, a = i)c_{1,i} + \sum_{i \in \mathcal{A}} \mathbb{I}(y < y_0, a = i)c_{0,i}.$$

We can compute that the expected loss under the forecasted distribution as follows.

$$\mathbb{E}_X \mathbb{E}_{\tilde{Y} \sim h[X]}[\ell(X, \tilde{Y}, \delta(X))] = \mathbb{E}_X \big[ \mathbb{I}(\delta(X) = 1)(\Pr(\tilde{Y} \geq y_0 \mid X)c_{11} + \Pr(\tilde{Y} < y_0 \mid X)c_{01})$$
$$+ \mathbb{I}(\delta(X) = 0)(\Pr(\tilde{Y} \geq y_0 \mid X)c_{10} + \Pr(\tilde{Y} < y_0 \mid X)c_{00})\big]$$

The decision maker minimizes their expected decision loss with respect to the forecasted distribution if $\delta(X) = 1$ when

$$(\Pr(\tilde{Y} \geq y_0 \mid X)c_{11} + \Pr(\tilde{Y} < y_0 \mid X)c_{01} \leq (\Pr(\tilde{Y} \geq y_0 \mid X)c_{10} + \Pr(\tilde{Y} < y_0 \mid X)c_{00}.$$

| Dataset | Method | Reliability Gap | Decision Loss | TCE | ECE |
|---|---|---|---|---|---|
| Protein ($n$=45730) | None | $0.175 \pm 0.012$ | $0.530 \pm 0.007$ | $0.123 \pm 0.007$ | $0.032 \pm 0.005$ |
| | Average | $0.172 \pm 0.005$ | $0.528 \pm 0.006$ | $0.104 \pm 0.007$ | $\mathbf{0.005 \pm 0.001}$ |
| | Threshold | $\mathbf{0.043 \pm 0.004}$ | $\mathbf{0.506 \pm 0.005}$ | $\mathbf{0.047 \pm 0.006}$ | $0.005 \pm 0.001$ |
| | Distribution | $0.055 \pm 0.024$ | $0.511 \pm 0.008$ | $0.058 \pm 0.010$ | $0.007 \pm 0.003$ |
| Energy ($n$=19735) | None | $0.092 \pm 0.021$ | $0.189 \pm 0.021$ | $0.178 \pm 0.030$ | $0.081 \pm 0.019$ |
| | Average | $0.055 \pm 0.007$ | $0.176 \pm 0.017$ | $0.071 \pm 0.003$ | $\mathbf{0.007 \pm 0.002}$ |
| | Threshold | $\mathbf{0.035 \pm 0.003}$ | $\mathbf{0.175 \pm 0.017}$ | $\mathbf{0.053 \pm 0.007}$ | $0.010 \pm 0.003$ |
| | Distribution | $0.063 \pm 0.013$ | $0.184 \pm 0.018$ | $0.121 \pm 0.015$ | $0.012 \pm 0.004$ |
| Naval ($n$=11934) | None | $0.085 \pm 0.006$ | $0.152 \pm 0.024$ | $0.245 \pm 0.031$ | $0.115 \pm 0.014$ |
| | Average | $0.103 \pm 0.016$ | $0.162 \pm 0.028$ | $0.068 \pm 0.011$ | $0.020 \pm 0.010$ |
| | Threshold | $\mathbf{0.038 \pm 0.006}$ | $\mathbf{0.126 \pm 0.024}$ | $\mathbf{0.047 \pm 0.006}$ | $\mathbf{0.013 \pm 0.003}$ |
| | Distribution | $0.046 \pm 0.007$ | $0.131 \pm 0.023$ | $0.071 \pm 0.011$ | $0.029 \pm 0.009$ |
| Crime ($n$=1994) | None | $0.101 \pm 0.008$ | $0.363 \pm 0.006$ | $0.125 \pm 0.010$ | $0.061 \pm 0.007$ |
| | Average | $0.081 \pm 0.016$ | $\mathbf{0.358 \pm 0.006}$ | $0.088 \pm 0.016$ | $\mathbf{0.019 \pm 0.008}$ |
| | Threshold | $\mathbf{0.070 \pm 0.011}$ | $\mathbf{0.358 \pm 0.006}$ | $\mathbf{0.073 \pm 0.017}$ | $0.021 \pm 0.009$ |
| | Distribution | $0.084 \pm 0.015$ | $0.378 \pm 0.007$ | $0.099 \pm 0.015$ | $0.027 \pm 0.009$ |

Table 5: Gaussian Forecaster Recalibration. Threshold calibration decreases the reliability gap. Despite that average calibration obtains low ECE, it can potentially increase the size of the reliability gap (see Naval dataset).

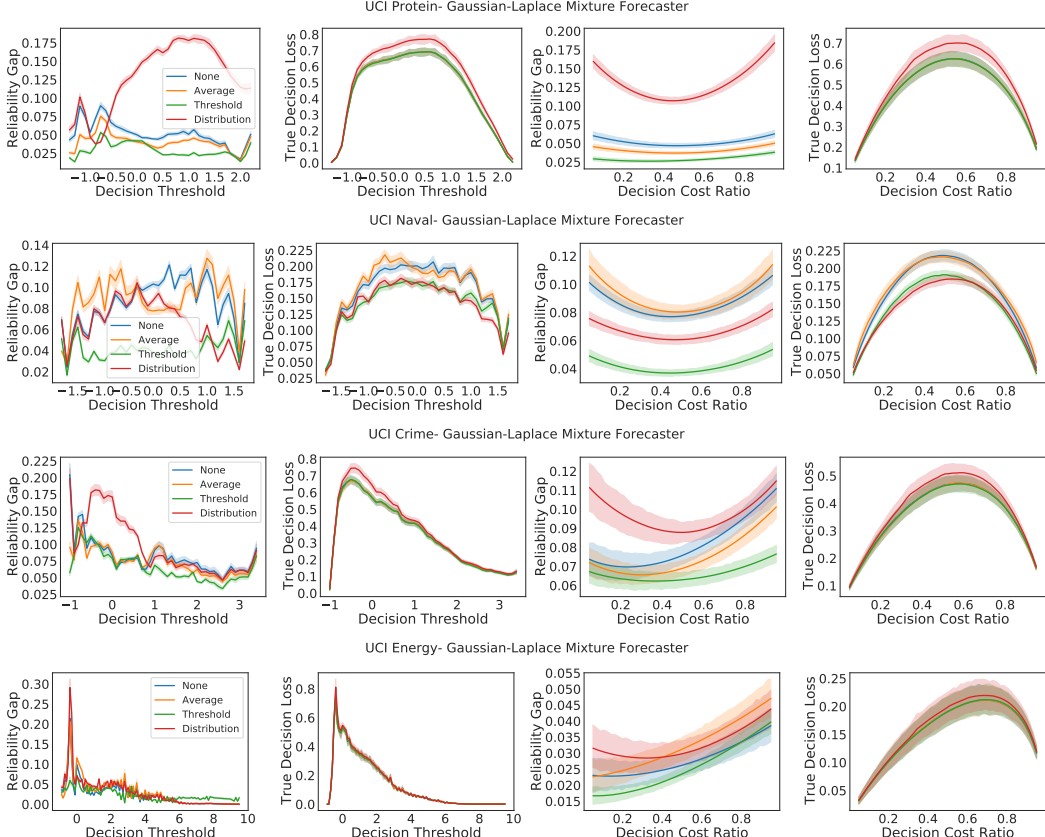

Figure 8: UCI Protein, Naval, Crime, Energy results with Gaussian-Laplace forecaster. As described in Section 5, the improvements from recalibration in the reliability gap and the decision loss are more modest because the uncalibrated forecaster that outputs Gaussian-Laplace distributions performs better than the uncalibrated forecaster that outputs Gaussian distributions. Nevertheless, threshold calibration offers improvement across decision thresholds and decision costs. In contrast, distribution calibration can enlarge the reliability gap and increase the decision loss.

| Dataset | Method | Reliability Gap | Decision Loss | TCE | ECE |
|---------|--------|-----------------|---------------|-----|-----|
| Protein | None | $0.052 \pm 0.009$ | $0.474 \pm 0.008$ | $0.054 \pm 0.009$ | $0.014 \pm 0.005$ |
| ($n$=45730) | Average | $0.041 \pm 0.006$ | $0.474 \pm 0.008$ | $0.046 \pm 0.008$ | $\mathbf{0.005 \pm 0.001}$ |
| | Threshold | $\mathbf{0.029 \pm 0.002}$ | $\mathbf{0.473 \pm 0.009}$ | $\mathbf{0.034 \pm 0.005}$ | $0.006 \pm 0.002$ |
| | Distribution | $0.130 \pm 0.025$ | $0.531 \pm 0.011$ | $0.113 \pm 0.011$ | $0.038 \pm 0.007$ |
| Energy | None | $0.028 \pm 0.007$ | $0.157 \pm 0.014$ | $0.075 \pm 0.011$ | $0.023 \pm 0.006$ |
| ($n$=19735) | Average | $0.029 \pm 0.008$ | $0.157 \pm 0.014$ | $0.058 \pm 0.015$ | $\mathbf{0.008 \pm 0.004}$ |
| | Point | $\mathbf{0.025 \pm 0.004}$ | $\mathbf{0.157 \pm 0.013}$ | $\mathbf{0.054 \pm 0.011}$ | $0.012 \pm 0.004$ |
| | Distribution | $0.033 \pm 0.007$ | $0.163 \pm 0.015$ | $0.089 \pm 0.016$ | $0.038 \pm 0.007$ |
| Naval | None | $0.086 \pm 0.009$ | $0.167 \pm 0.012$ | $0.202 \pm 0.075$ | $0.091 \pm 0.039$ |
| ($n$=11934) | Average | $0.091 \pm 0.017$ | $0.170 \pm 0.012$ | $0.083 \pm 0.021$ | $0.014 \pm 0.007$ |
| | Threshold | $\mathbf{0.042 \pm 0.009}$ | $0.145 \pm 0.011$ | $\mathbf{0.055 \pm 0.012}$ | $\mathbf{0.013 \pm 0.007}$ |
| | Distribution | $0.067 \pm 0.014$ | $\mathbf{0.143 \pm 0.013}$ | $0.183 \pm 0.031$ | $0.086 \pm 0.017$ |
| Crime | None | $0.081 \pm 0.017$ | $0.357 \pm 0.014$ | $0.091 \pm 0.024$ | $0.030 \pm 0.010$ |
| ($n$=1994) | Average | $0.075 \pm 0.020$ | $0.356 \pm 0.015$ | $0.079 \pm 0.026$ | $\mathbf{0.022 \pm 0.007}$ |
| | Threshold | $\mathbf{0.066 \pm 0.010}$ | $\mathbf{0.355 \pm 0.010}$ | $\mathbf{0.078 \pm 0.016}$ | $0.026 \pm 0.007$ |
| | Distribution | $0.097 \pm 0.018$ | $0.382 \pm 0.021$ | $0.093 \pm 0.019$ | $0.034 \pm 0.012$ |

Table 6: Gaussian-Laplace Mixture Forecaster Recalibration. Threshold calibration can decrease the size of the reliability gap. Distribution calibration can be challenging to enforce in model families with more parameters, and we see that it can be detrimental to the performance of the forecaster.

The Bayes decision rule with respect to the forecasted distribution is

$$\delta^*(X) = \begin{cases} 1 & \text{if } \Pr(\tilde{Y} < y_0 \mid X) \leq \frac{c_{01} - c_{11}}{c_{01} + c_{10} - c_{11} - c_{00}} \\ 0 & \text{else} \end{cases}$$

Equivalently,

$$\delta^*(X) = \begin{cases} 1 & \text{if } h[X](y_0) \leq \frac{c_{01} - c_{11}}{c_{01} + c_{10} - c_{11} - c_{00}} \\ 0 & \text{else} \end{cases}.$$

Thus, the Bayes decision rule is a threshold decision rule given by

$$\delta^*(X) = \mathbb{I}\Big(h[X](y_0) \leq \frac{c_{01} - c_{11}}{c_{01} + c_{10} - c_{11} - c_{00}}\Big).$$

## B.2 Proof of Lemma 1

*Proof.* Let $f$ be a forecaster $h : \mathcal{X} \to \mathcal{F}(\mathcal{Y})$ that satisfies Definition 3. Then we have that

$$\Pr[h[X](Y) \leq c] = \Pr[h[X](Y) \leq c \mid h[X](y_0) \leq 1] = c, \tag{2}$$

where $y_0 \in \mathcal{Y}$. By the law of total probability, we have that

$$\Pr[h[X](Y) \leq c] = \Pr[h[X](Y) \leq c, h[X](y_0) > \alpha] + \Pr[h[X](Y) \leq c, h[X](y_0) \leq \alpha]. \tag{3}$$

From Definition 3, we have that $\alpha \in [0, 1], y_0 \in \mathcal{Y}$,

$$\Pr[h[X](y_0) \leq c \mid h[X](y_0) \leq \alpha] = c.$$
$$\frac{\Pr[h[X](y_0) \leq c, h[X](y_0) \leq \alpha]}{\Pr[h[X](y_0) \leq \alpha]} = c$$
$$\frac{Pr[h[X](Y) \leq c] - \Pr[h[X](Y) \leq c, h[X](y_0) > \alpha]}{1 - \Pr[h[X](y_0) > \alpha)} = c$$
$$\frac{c - \Pr[h[X](Y) \leq c, h[X](y_0) > \alpha]}{1 - \Pr[h[X](y_0) > \alpha)} = c$$

Rearranging the terms, we find that

$$\Pr[h[X](Y) \le c \mid h[X](y_0) > \alpha] = \frac{\Pr[h[X](Y) \le c, h[X](y_0) > \alpha]}{\Pr[h[X](y_0) > \alpha]} = c.$$

Thus, if a forecaster satisfies Definition 3, then it also satisfies

$$\Pr[h[X](Y) \le c \mid h[X](y_0) > \alpha] = c \quad \forall y_0 \in \mathcal{Y}, \forall \alpha \in [0,1], c \in [0,1].$$

$\square$

### B.3 Proof of Theorem 1

*Proof of Theorem 1.* Before proving the theorem we need a simple Lemma

**Lemma 2.** *For any pair of random variables $U, V$, $\mathbb{E}[U \mid V] = 0$ almost surely if and only if $\forall c \in \mathbb{R}, \mathbb{E}[U\mathbb{I}(V > c)] = 0$.*

We first show that if a forecaster $h$ is threshold-calibrated, then the forecaster yields zero reliability gap for any threshold decision rule under any threshold loss function. Suppose we have a threshold loss function with decision threshold $y_0 \in \mathcal{Y}$.

Let $U = h^*[X](y_0)$ and $\tilde{U} = h[X](y_0)$. Suppose $h$ satisfies threshold calibration $\Pr[h[X](Y) \le c \mid h[X](y_0) \le \alpha] = c$, under the new notation this implies that

$$\mathbb{E}[\mathbb{I}(h[X](Y) \le c) - c \mid \tilde{U} \le \alpha] = 0$$

We can further derive

$$
\begin{aligned}
\mathbb{E}[U - \tilde{U} \mid \tilde{U} \le \alpha] &= \mathbb{E}[h^*[X](y_0) - h[X](y_0) \mid \tilde{U} \le \alpha] \\
&= \mathbb{E}[\mathbb{I}(Y \le y_0) - h[X](y_0) \mid \tilde{U} \le \alpha] \\
&= \mathbb{E}[\mathbb{I}(h[X](Y) \le h[X](y_0)) - h[X](y_0) \mid \tilde{U} \le \alpha] \quad \text{Monotonicity} \\
&= 0
\end{aligned}
$$

Therefore, we have that $\mathbb{E}[(U - \tilde{U})\mathbb{I}(\tilde{U} \le \alpha)] = 0, \forall \alpha \in [0,1]$. We can use this fact to show that the reliability gap must be equal to $0$.

For any loss function $\ell$ and threshold decision rule $\delta_h$ we have that the true average decision loss can be written as follows:

$$\mathbb{E}[\ell(X, Y, \delta_h(X))] = \mathbb{E}[\ell(X, Y, 1)\mathbb{I}(\delta_h(X) = 1)] + \mathbb{E}[\ell(X, Y, 0)\mathbb{I}(\delta_h(X) = 0)]$$

Similarly, the predicted average decision loss can be written as follows:

$$\mathbb{E}[\ell(X, \tilde{Y}, \delta_h(X))] = \mathbb{E}[\ell(X, Y, 1)\mathbb{I}(\delta_h(X) = 1)] + \mathbb{E}[\ell(X, Y, 0)\mathbb{I}(\delta_h(X) = 0)]$$

WLOG, it suffices to show that

$$\mathbb{E}[\ell(X, Y, 1)\mathbb{I}(\delta_h(X) = 1)] - \mathbb{E}[\ell(X, \tilde{Y}, 1)\mathbb{I}(\delta_h(X) = 1)] = 0 \quad \forall \alpha, c \in [0,1], y_0 \in \mathcal{Y}.$$

We find that

$$
\begin{aligned}
&\mathbb{E}[\ell(X, Y, 1)\mathbb{I}(\delta_h(X) = 1)] - \mathbb{E}[\ell(X, \tilde{Y}, 1)\mathbb{I}(\delta_h(X) = 1)] \\
=&\mathbb{E}[c_{11}\mathbb{I}(Y \ge y_0, \delta_h(X) = 1)] + \mathbb{E}[c_{01}\mathbb{I}(Y \le y_0, \delta_h(X) = 1)] \\
&- \mathbb{E}[c_{11}\mathbb{I}(\tilde{Y} \ge y_0, \delta_h(X) = 1)] - \mathbb{E}[c_{01}\mathbb{I}(\tilde{Y} < y_0, \delta_h(X) = 1)] \\
=&\mathbb{E}[c_{11}(1 - \mathbb{I}(Y < y_0))\mathbb{I}(\delta_h(X) = 1)] + \mathbb{E}[c_{01}\mathbb{I}(Y < y_0)\mathbb{I}(\delta_h(X) = 1)] \\
&- \mathbb{E}[c_{11}(1 - \mathbb{I}(\tilde{Y} < y_0)\mathbb{I}(\delta_h(X) = 1)] - \mathbb{E}[c_{01}\mathbb{I}(\tilde{Y} < y_0)\mathbb{I}(\delta_h(X) = 1)] \\
=&\mathbb{E}[c_{11}(1 - U)\mathbb{I}(\tilde{U} \le \alpha)] - \mathbb{E}[c_{01}U\mathbb{I}(\tilde{U} \le \alpha)] \\
&- \mathbb{E}[c_{11}(1 - \tilde{U})\mathbb{I}(\tilde{U} \le \alpha)] - \mathbb{E}[c_{01}U\mathbb{I}(\tilde{U} \le \alpha)] \\
=&0.
\end{aligned}
$$

The first line follows from the definition of $\ell$. The second line holds because $\mathbb{I}(Y \geq y_0) = 1 - \mathbb{I}(Y < y_0)$. The thirdline follows from the definition of $U$ and $\tilde{U}$. The last line follows from the fact that $\mathbb{E}[U - \tilde{U} \mid \tilde{U}] = 0$ almost surely. Thus, if $h$ is threshold calibrated, then the reliability gap is equal to zero.

We also show that the converse holds. If for any threshold loss $\ell$ and threshold decision rule $\delta_h$ we have

$$\mathbb{E}[\ell(X, Y, \delta_h(X))] - \mathbb{E}[\ell(X, \tilde{Y}, \delta_h(X))] = 0,$$

then $\mathbb{E}[(U - \tilde{U})\mathbb{I}(\tilde{U} \leq \alpha)] = 0$ for any $\alpha \in [0, 1]$. As a result, by Lemma 2 $\mathbb{E}[U - \tilde{U} \mid \tilde{U}] = 0$ almost surely, so we have that

$$\Pr[h[X](Y) \leq c \mid h[X](y_0) = \alpha] = c \quad \forall \alpha, c \in [0, 1], y_0 \in \mathcal{Y}$$

which is equivalent to the threshold calibration condition:

$$\Pr[h[X](Y) \leq c \mid h[X](y_0) \leq \alpha] = c \quad \forall \alpha, c \in [0, 1], y_0 \in \mathcal{Y}.$$

$\square$

### B.4 Proof of Proposition 1

*Proof.* Let $h$ be a forecaster $h : \mathcal{X} \to \mathcal{F}(\mathcal{Y})$ where $\mathcal{F}(\mathcal{Y})$ is a class of continuous CDFs mapping $\mathcal{Y} \to [0, 1]$.

1. Suppose a forecaster $h$ satisfies distribution calibration, we show that it must also be threshold-calibrated. Let $g \in \mathcal{F}(\mathcal{Y})$. For any $y_0 \in \mathcal{Y}$ and $\alpha \in [0, 1]$,

$$
\begin{aligned}
\Pr[h[X](Y) \leq c \mid h[X](y_0) \leq \alpha] &= \mathbb{E}[\mathbb{I}(h[X](Y) \leq c) \mid h[X](y_0) \leq \alpha] \\
&= \mathbb{E}[\Pr[h[X](Y) \leq c \mid h[X](y_0) \leq \alpha, h[X] = g]] \\
&= \mathbb{E}[\Pr[h[X](Y) \leq c \mid g(y_0) \leq \alpha, h[X] = g]] \\
&= \mathbb{E}[\Pr[h[X](Y) \mid h[X] = g]] \\
&= c.
\end{aligned}
$$

The first line is from the definition of probability. The second is due to law of iterated expectations. The third line also follows from the fact that $h[X] = g$ determines whether $h[X](y_0) \leq \alpha$. The third line follows from the definition of distribution calibration.

2. Suppose a forecaster $h$ satisfies threshold calibration, then it must be average-calibrated. For all $x \in \mathcal{X}, y_0 \in \mathcal{Y}, \alpha = 1$

$$
\begin{aligned}
\Pr[h[X](Y) \leq c] &= \Pr[h[X](Y) \leq c \mid h[X](y_0) \leq 1] \\
&= c.
\end{aligned}
$$

$\square$

### B.5 Proof of Theorem 2

*Proof of Theorem 2.* Denote $\mathcal{X}_0 = \{x, h^t[x](y_t) \leq \alpha_t\}$ and $\mathcal{X}_1 = \{x, h^t[x](y_t) > \alpha_t\}$ and suppose that

$$\sum_{j=0,1} P(\mathcal{X}_j) \int_y (F_{Y|\mathcal{X}_1}(y) - F^*_{Y|\mathcal{X}_j}(y))^2 dy \geq \epsilon$$

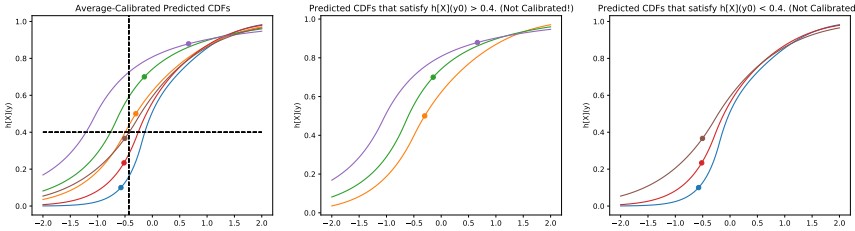

Figure 9: We partition the predicted CDFs into two groups 1) $h[X](y_0) > \alpha$, 2) $h[X](y_0) < \alpha$. The dot on each CDF denotes the location of the true label. **Left**: We observe that $h[X](Y)$ is (approximately) uniformly distributed, so the predicted CDFs satisfy average calibration. Dashed lines denote the position of $y_0$ and $\alpha$. **Middle:** The forecaster is not average calibrated conditioned on the subset of predicted CDFs where $h[X](y_0) > \alpha$ because $h[X](Y)$ is not uniformly distributed. **Right**: The forecaster is not average calibrated conditioned on the predicted CDFs where $h[X](y_0) < \alpha$ because $h[X](Y)$ is not uniformly distributed in this part. As a result, average calibration does not imply threshold calibration.

Choose as the potential $\mathbb{E}\left[\left|\int_y F^*[X](y) - \tilde{F}_t[X](y)\right|^2\right]$, denote $\gamma_j(y) = F^*_{Y|\mathcal{X}_j}(y) - \tilde{F}_{Y|\mathcal{X}_j}(y)$ then

$$\mathbb{E}\left[\int_y (F^*[X](y) - \tilde{F}_t[X](y))^2\right] - \mathbb{E}\left[\int_y (F^*[X](y) - \tilde{F}_{t+1}[X](y))^2\right]$$

$$= \sum_j P(\mathcal{X}_j)\mathbb{E}\left[\int_y (F^*[X](y) - \tilde{F}_t[X](y))^2 - (F^*[X](y) - \tilde{F}_{t+1}[X](y))^2 \mid \mathcal{X}_j\right] \qquad \text{Tower}$$

$$= \sum_j P(\mathcal{X}_j)\mathbb{E}\left[\int_y (F^*[X](y) - \tilde{F}_t[X](y))^2 - (F^*[X](y) - \tilde{F}_t[X](y) - \gamma_j(y))^2 \mid \mathcal{X}_j\right]$$

$$= \sum_j P(\mathcal{X}_j)\mathbb{E}\left[\int_y (2F^*[X](y) - 2\tilde{F}_t[X](y) - \gamma_j(y))\gamma_j(y) \mid \mathcal{X}_j\right]$$

$$= \sum_j P(\mathcal{X}_j)\int_y \gamma_j(y)^2 \geq \epsilon$$

$\square$

## C   Counterexample

In Figure 9, we give a visualization of why average calibration does not necessarily imply threshold calibration. Although the forecaster satisfies average calibration across the predicted CDFs (leftmost plot), we notice that the forecaster is not calibrated when conditioned on subsets of the predicted CDFs that satisfy $h[X](y_0) \leq \alpha$ and $h[X](y_0) > \alpha$, respectively.