# OpenReview forum: "Reliable Decisions with Threshold Calibration"
_NeurIPS.cc/2021/Conference — NeurIPS 2021 Poster_

### Official Review · Reviewer_YN6e · 2021-07-15

**Rating:** 8
**Confidence:** 4

**Summary:**

The paper proposes a new concept called threshold calibration of probabilistic regression models, and proposes a method with theoretical calibration guarantees. The method is evaluated on real datasets and is shown to outperform existing methods.


**Limitations And Societal Impact:**

The limitations and social impact have been covered adequately.

**Main Review:**

The paper fills an important gap between average calibration and distribution calibration. The ideas are simple and well justified, the intuitions have been explained well, a technically strong paper. I have only the following minor comments.

From the definition of the reliability gap it is not obvious whether the gap could also become negative. I suggest to
 have an answer to this question either in the main paper or in the appendices.

All the brackets have the same size in all the formulas, I recommend using different sizes. In formulas like the one
in Definition 4 it would be much easier to visually grasp where the subexpressions start and end. However, it is cert
ainly also a matter of taste.

In lines 182-188 on page 5 the claims (average calibration does not imply threshold calibration does not imply distribution calibration) are intuitively obvious, but perhaps particular examples would be useful (could also be presented in the appendix).

Algorithm 1 refers to the use of isotonic regression, but some additional details about the implementation should be mentioned. The isotonic function is uniquely determined on its training points, but it could be interpolated in various ways. One possible choice is the constant interpolation (e.g. each test point gets the same label as the closest training point that is less or equal to this test point), and another choice is linear interpolation.

Line 261 states that $\mathcal{Y}_0$ has 50 uniformly spaced points spanning the label space. It would be useful to mention that it is spanning the label space between its minimal and maximal value in the training (?) data set.

The sentence 'Let $\mathcal{C}$ be the set of possible choices for $\alpha$.' in page 7, line 262 could be rephrased because the previous sentence has basically already stated the same.

The notation $\delta^i_{h,\alpha_*}$ on line 263 and $\delta^i_{h,\alpha^*}$ on line 264 is confusing. If I understand correctly, then the asterisk should really be in the superscript of $\delta$ in accordance to Definition 1 of the Bayes decision rule.

==
After discussion:
I am happy to stay at my suggested grade 8.


**Time Spent Reviewing:**

4

---

> ### Author Response · Authors · 2021-08-11
> **Clarifications on the reliability gap, counter-example, experiments.**
>
> Thank you for your comments and suggestions for our paper.
>
> $Q$: Can the decision gap be negative?
>
> The difference between the true decision loss and the decision loss predicted by the forecaster can be positive or negative. In Figure 1, we show that an average-calibrated forecaster can overestimate the true decision loss in some regions and underestimate the true decision loss in others.
> The reliability gap is defined to be the absolute difference between the observed decision loss and the decision loss predicted by the model, so the reliability gap is nonnegative (as depicted in Figure 1).
>
> $Q$: Counter-example that shows average calibration may not imply threshold calibration.
>
> We can add the following illustration to the appendix of the text. This is an illustration that shows that average calibration may not necessarily imply threshold calibration. https://drive.google.com/file/d/1qur6YsJDKvRz7gytyB4Sb3aNCr0rEqyc/view?usp=sharing
>
> $Q$: Choice of interpolation in isotonic regression.
>
> We use linear interpolation, which is the default interpolation in Scipy’s 1D interpolation method (https://docs.scipy.org/doc/scipy/reference/generated/scipy.interpolate.interp1d.html).
>
> $Q$: Writing suggestions.
>
> Thank you for these comments. We have updated our paper with these changes.

---

### Official Review · Reviewer_Y7ES · 2021-07-18

**Rating:** 5
**Confidence:** 3

**Summary:**

As pointed out by the paper, threshold calibration is related with practical hospital scheduling problem, in which hospital needs to make the decision to acquire new patients based on predicted length-of-stay to the existing patients. The paper proposed an efficient and practical threshold calibration method, which calibrated the non-calibrated results based on CDF prediction, and is shown to have better calibration results as compared to average calibration and distribution calibration.  Additionally, the paper proposed an algorithm to do threshold calibration, and empirical results on real data sets shows reduced reliability gap and true decision loss.

**Ethics Review Area:**

["I don’t know"]

**Limitations And Societal Impact:**

Not Applied

**Main Review:**

[1] The paper formulates the problems into regression problems, predicts the likelihood of stay less or equal than y_0 and makes decision based on the if the likelihood is above a threshold \alpha, as mentioned in section 2.3. How are \alpha and y_0 get selected, are they flexible to tune or pre-determined? Since the problem is mainly to predict the decision, what is the major reason to formulate the problem as a regression problem rather than a classification problem?

[2] This paper proposes a min-max approach to find y_0, \alpha to maximize TCE, followed by a greedy approach to fix the mis-calibration with the selected breakdown identified by y_0 and \alpha, as described in algorithm 1. How to select y_0 and alpha to maximize TCE, is it via grid search?

**Time Spent Reviewing:**

2

---

> ### Author Response · Authors · 2021-08-11
> **Clarifications on problem setup and algorithm**
>
> Thank you for your comments and feedback on our paper.
>
> $Q$: How are $\alpha$ and $y_{0}$ selected?
>
> Decision makers can select the values $y_{0}$ and $\alpha$ (based on their loss function). However, our recalibration algorithm achieves threshold calibration for **all** $y_0$ and $\alpha$. The provider of a machine learning model might not know the threshold decision task (i.e. values of $y_{0}$ and $\alpha$) a downstream decision maker may be interested in. To accommodate such scenarios, we calibrate the model for any threshold decision task (any choice of $y_{0}$ and $\alpha$).
>
> $Q$: Why formulate the problem as regression instead of classification?
>
> The reason is reusability. The same prediction model might be used by multiple decision makers with different decision losses. For example, medical prediction models are often deployed in many different hospitals with different decision tasks. If there is a single decision task, then we can transform the problem into a classification task (i.e. predict the optimal action), but we lose the reusability and must train a separate model for each decision task.
>
> $Q$: How are $y_{0}, \alpha$ selected to maximize TCE, is it via grid search?
>
> Yes, we find $y_{0}$ and $\alpha$ via grid search. We choose $y_0$ from a discretization of the range of the validation labels and $\alpha$ from a discretization of the interval $[0, 1]$.

---

### Official Review · Reviewer_H5Nm · 2021-07-18

**Rating:** 8
**Confidence:** 4

**Summary:**

This work applies the concept of calibrated regression into a decision making framework where the decision is confined to threshold decisions. The Bayes optimal decision w.r.t. the predicted distribution is derived, and the concept of threshold calibration is defined. Threshold calibration is a sufficient and necessary condition for the reliability gap to be 0, meaning the absolute difference between the predicted expected loss and the true expected loss is 0 for a given decision rule. Further, a simple algorithm is proposed which threshold recalibrates a distributional prediction.


**Limitations And Societal Impact:**

The limitations are well addressed in Section 7, with constructive statements about future directions for improvements.

**Main Review:**

This paper is very clearly written and the topics/problems discussed are well motivated. While there is a lot recent attention on achieving calibration for distributional predictions, exactly $\textit{how}$ calibration can be used in a downstream task and to what degree it is desirable has not been discussed very much, and this paper provides a thorough analysis at least in the context of threshold decisions.
The newly defined concepts are also very relevant and theoretically sound.
While the empirical results are not especially surprising given threshold recalibration is tailored to the problem setting and evaluation metrics used (reliability gap), it does provide a good baseline for future works in applications of calibration in decision making tasks.
I believe the problem setting, concepts, derivations, and methods this work provides will be of value to the general UQ community.

Just a couple of questions/points:
1. In many of the definitions, while I understand what the authors mean, I believe the notation is slightly off. E.g. in average calibration (Definition 5, L170), the statement should not be for small $x$, but for the random variable $X$. Small $x$ is a realized value (as the authors also indicate in the Preliminaries), which implies conditioning each $x \in \mathcal{X}$. In this case, it would indicate another notion, "individual calibration" [33].
The same point goes for other places too, e.g. Definition 6.
I am aware the notation is correctly written in various places in the Appendix, e.g. L572.

2. Was any hyperparameter tuning performed for the experiments? I cannot find any details in the Appendix. While distribution calibration is reported to degrade reliability gap (L293), can this simply be because there are not enough points in the recalibration set (as the authors also hypothesize)? I'm wondering if simply tuning the percentage of validation points used for recalibration (currently set to 10% of training) can alleviate this.

3. Lastly, the code link in the Appendix doesn't work.


**Time Spent Reviewing:**

5

---

> ### Author Response · Authors · 2021-08-11
> **Clarification on notation, hyperparameter tuning, and code.**
>
> Thank you for your comments and suggestions on our paper. We appreciate your positive comments on the significance of our work and relevance to the UQ community.
>
> $Q$: The notation is slightly off.
>
> We acknowledge an unfortunate typo in Definition 3-6 and Lemma 1 where we miswrote the capital X (which traditionally denotes a random variable) as lower case x (which traditionally denotes a value). For example, Eq.(1) in definition 3 should instead read
> $$
> P[ h[ X ] ( Y ) \leq c \mid h[ X ] (y_{0}) \leq \alpha ] = c \quad \forall y_{0} \in \mathcal{Y}, \alpha \in [0, 1], \forall c \in [0, 1].
> $$
> As you mention, in the proofs and the Appendix, our notation follows the correct convention.
>
> $Q$: Hyperparameter tuning on validation set size.
>
> We did not perform hyperparameter tuning on the size of the validation set. We expect that increasing the number of samples will improve the performance of both threshold calibration and distribution calibration.
>
> $Q$: The code link does not work.
>
> The code is available here: https://drive.google.com/file/d/12Qh1AWsJcx6UzrRAVYPAYPNemRBj7500/view?usp=sharing

---

> > ### Comment · Reviewer_H5Nm · 2021-09-10
> > **Thanks for the clarifications**
> >
> > Thank you for the clarifications.
> > As mentioned by other reviewers, I think one possible way to strengthen this work could be to include an ablation study on the effect of the recalibration dataset size, which is currently fixed.
> >
> > Otherwise, after reading the submission again and other reviews+responses, I believe there are many positive aspects of this work and I am happy to raise my score to 8.

---

### Official Review · Reviewer_CBqV · 2021-07-18

**Rating:** 4
**Confidence:** 4

**Summary:**

This paper studies calibrated prediction from a decision loss perspective. The main contribution is the proposal of a new notion of threshold calibration, an algorithm to learn predictors achieving this new notion of calibration, and well as experiments demonstrating the effectiveness of the proposed algorithm.

**Limitations And Societal Impact:**

yes, the authors address it

**Main Review:**

This work studies an important problem of learning regression classifiers which are well calibrated for the purpose of downstream decision-making. While the setup is interesting, there are few points worth highlighting.
- Definition 3 which introduces the notion of threshold calibration is quite opaque the way it is written. I would like to see some intuition into why this should be the correct notion (I can see from the proof of Theorem 1 how it arises, but am still unable to get an intuitive idea of this definition).
- Proof sketch is missing (Theorem 1). While I understand that there is a page limit on submission, I think it is very important to highlight to both the reviewers and readers that the theorem is provably correct, either by providing some intuition or giving a sketch of the main idea.
- One can ask this question of decision based calibration in a much more general framework (rather than just focussing on threshold functions for decisions). Can the authors perhaps point out how the notion of calibration change for general functions mapping cdfs to decisions.
- On line 182, the authors mention that the converse of Proposition 1 is not true. Can the authors perhaps add in a proof of this to the paper, giving counter examples to the two statements.
- Theorem 2: What happens to the prediction accuracy of the forecaster h^T as compared to the input forecaster h, when evaluated on the regression prediction task? What about sample complexity issues? How many samples are required to perform this recalibration -- this is quite important since I feel that the notion of threshold calibration as defined for all (x, y, c) tuple is indeed quite a strong one.
- The experiment section of the paper is quite weak as well. I would like to see some some more simulation experiments which are directed towards studying the limitations of the proposed methods, some more loss functions l^i, a wider variety of decision rules, different choices of recalibration functions (based on quantile methods, optimizing some variant of the pinball loss) as well as additional baselines (optimizing directly for the decision loss, fine tuning the regression function for the decision loss).


**Time Spent Reviewing:**

2

---

> ### Author Response · Authors · 2021-08-11
> **Clarification on intuition for definitions and theorems and on algorithm.**
>
>
> Thank you for your comments and suggestions on our paper.
>
> $Q$: Intuition on the definition of threshold calibration.
>
> The intuition is that if we partition the samples into two subsets, one set with predicted CDFs that satisfy $h[ X ] (y_0) < \alpha$ and the other with predicted CDFs that satisfy $h[ X ] ( y_0 ) \geq \alpha$, then both sets of the partition will achieve average calibration. Another way to think of this is that in a decision task, a (Bayes) decision maker chooses $a=0$ and $a=1$ respectively for each subset (This is proven in Appendix B.1). We are enforcing (average) calibration for the subset where the decision maker chooses $a=0$ and $a=1$ respectively.
>
>
> $Q$: Intuition/sketch for Theorem 1 in the main paper.
>
> The expected decision loss under the true distribution can be decomposed into two terms. The first term corresponds to the cost incurred from “false positive” errors and the second term corresponds to the cost incurred from “false negative” errors. Under threshold calibration, the forecaster’s predicted error rates match the true error rates. Since the decision loss (with any choice of costs) is a linear combination of these error rates, the expected decision loss predicted by the forecaster matches the expected decision loss under the true distribution.
>
> $Q$: General framework for decision-based calibration beyond threshold decision rules.
>
> The motivation and the reliability gap we defined will still hold for any set of decision loss functions. If the decision loss is not a threshold loss however, to obtain any guarantees on the decision loss gap, the recalibration algorithm will have to be revised. Nevertheless we highlight that our recalibration algorithm should not hurt performance even when the decision loss is not a threshold function. Our recalibration algorithm reduces the L2 error between the predicted CDF and the true CDF (which is how we proved Theorem 1).
>
> $Q$: Counter-examples that show average calibration does not imply threshold calibration.
>
> We can add the following illustration to the appendix of the text. This is an illustration that shows that average calibration does not necessarily imply threshold calibration. https://drive.google.com/file/d/1qur6YsJDKvRz7gytyB4Sb3aNCr0rEqyc/view?usp=sharing
>
>
> $Q$: Predictive accuracy of forecaster after calibration algorithm.
>
> The accuracy of the forecaster should not be harmed. In fact, the L2 error between the predicted CDF and the true CDF decreases by at least $\epsilon$ after each recalibration step --- this is how we prove convergence in Theorem 1.
>
> $Q$: Sample complexity of algorithm.
>
> Theoretical sample complexity bounds are usually loose, and practical sample complexity bounds usually come from empirical evaluation with held out data. We take such a strategy in the paper. Empirically, our smallest recalibration set contains 835 samples, but we observe significantly improved TCE on the held out test set (from an error 0.12 with the original predictor to 0.07 with our algorithm).

---

### Decision · Program_Chairs · 2021-09-28

**Decision:**

Accept (Poster)

**Comment:**

This paper looks at how to guarantee accurate decision loss prediction when the predictions are based on machine learning models. The authors propose the notion of "threshold calibration" which provides a stronger guarantee than the traditional average notion of calibration. This ensures that decision loss is predicted accurately when the threshold decision is used.

A threshold calibration is a promising approach compared to average calibration or distribution calibration. Reviewer H5Nm and YN6e concur that the results presented in this paper "fill an important gap between average calibration and distribution calibration". However, Reviewer CBqV raises a point that the empirical evaluation could be extended further, hence recommending a low score. Reviewer Y7ES thought the paper is borderline and did not provide a strong justification in favor of one way or another. Hence, it does not play an important role in my decision.

I find Reviewer CBqV's argument on the empirical evaluation relatively weak, especially for the venues like NeurIPS, and after reading the paper, I also find the studied problem to be quite timely. The paper is well-written as well as thought-provoking. The only suggestion I have for the authors is to move the related work section toward the beginning of the paper. This could help the non-expert readers better understand the literature.

Following the recommendation of Reviewer H5Nm and YN6e, I recommend this paper for publication at NeurIPS2021 as a poster.

**Consistency Experiment:**

NeurIPS has a long history of experimentation. In 2014, NeurIPS ran an experiment in which 10% of submissions were reviewed by two independent committees to quantify the randomness in the review process. This year, we repeated a variant of this experiment to see how the quality of the review process has changed over time.  This paper was part of the experiment and was therefore assigned to two committees (consisting of reviewers, an Area Chair, and a Senior Area Chair) that reached independent decisions.  If both committees made the same recommendation, this recommendation was followed. If a single committee recommended acceptance, the paper was accepted (with the exception of a few cases in which the other committee identified what we considered a fatal flaw, e.g., an error in a key result).

Both committees reached the same decision: **Accept (Poster)**

The other committee assigned to the paper recommended **Accept (Poster)**.  You can find the other set of reviews, along with any follow up discussion with the authors here:
https://openreview.net/forum?id=Mx-iNoxLU4t